# Specificity, propagation, and memory of pericentric heterochromatin

Katharina Müller-Ott[1], Fabian Erdel[1], Anna Matveeva[2], Jan-Philipp Mallm[1], Anne Rademacher[1], Matthias Hahn[3], Caroline Bauer[1], Qin Zhang[2], Sabine Kaltofen[1,†], Gunnar Schotta[3], Thomas Höfer[2] & Karsten Rippe[1,*]

## Abstract

The cell establishes heritable patterns of active and silenced chromatin via interacting factors that set, remove, and read epigenetic marks. To understand how the underlying networks operate, we have dissected transcriptional silencing in pericentric heterochromatin (PCH) of mouse fibroblasts. We assembled a quantitative map for the abundance and interactions of 16 factors related to PCH in living cells and found that stably bound complexes of the histone methyltransferase SUV39H1/2 demarcate the PCH state. From the experimental data, we developed a predictive mathematical model that explains how chromatin-bound SUV39H1/2 complexes act as nucleation sites and propagate a spatially confined PCH domain with elevated histone H3 lysine 9 trimethylation levels via chromatin dynamics. This "nucleation and looping" mechanism is particularly robust toward transient perturbations and stably maintains the PCH state. These features make it an attractive model for establishing functional epigenetic domains throughout the genome based on the localized immobilization of chromatin-modifying enzymes.

**Keywords** FRAP/FCS; heterochromatin protein 1; histone methylation; pericentric heterochromatin; protein network
**Subject Categories** Quantitative Biology & Dynamical Systems; Chromatin, Epigenetics, Genomics & Functional Genomics
**Mol Syst Biol. (2014) 10: 746**

## Introduction

Epigenetic networks control the accessibility of DNA for transcription, DNA repair, and replication machineries. They establish and maintain different functional chromatin states through cell division via protein factors that set or remove specific modifications of histones and DNA in the absence of alterations of the DNA sequence

(Berger *et al*, 2009). These chromatin signals in turn recruit architectural chromatin components or chromatin remodeling factors in a highly dynamic manner and regulate genome access (McBryant *et al*, 2006; Taverna *et al*, 2007; Campos & Reinberg, 2009; Clapier & Cairns, 2009; Erdel *et al*, 2011a). On a global scale, the concerted and targeted activity of these networks results in the formation of the denser, transcriptionally repressed heterochromatin state and the more open and biologically active euchromatin, which can be distinguished at the resolution of the light microscope (Grewal & Jia, 2007; Eissenberg & Reuter, 2009). A prototypic example for a constitutive heterochromatin domain is pericentric heterochromatin (PCH) in mouse cells (Probst & Almouzni, 2008). It is characterized by its high content of repetitive major satellite repeats and repressive epigenetic marks such as 5-methylcytosine (5meC) the binding of proteins with a methyl-CpG-binding domain that recognize this modification, trimethylation of histone H3 lysine residue 9 (H3K9me3), and histone H4 lysine residue 20 (H4K20me3), as well as hypoacetylation of histones (Probst & Almouzni, 2008). The H3K9me3 modification is set by the histone methylases SUV39H1 and SUV39H2 (in the following "SUV39H" refers to both isoforms), while SUV4-20H1 and SUV4-20H2 set H4K20me2 and promote H4K20me3 ("SUV4-20H" for both isoforms) (Kwon & Workman, 2008; Schotta *et al*, 2008; Eissenberg & Reuter, 2009; Byrum *et al*, 2013).

A central protein component of PCH is heterochromatin protein 1 (HP1) that is present in three very similar isoforms HP1α, HP1β, and HP1γ in mice and humans (Maison & Almouzni, 2004; Hiragami & Festenstein, 2005; Kwon & Workman, 2008). HP1 contains an N-terminal chromodomain (CD) and a C-terminal chromoshadow-domain (CSD) connected by a flexible linker region. The CD interacts specifically with H3 histone tails that carry the K9me3 modification (Jacobs & Khorasanizadeh, 2002; Fischle *et al*, 2003). HP1 is able to form homo- and heterodimers (Nielsen *et al*, 2001; Yamamoto & Sonoda, 2003; Rosnoblet *et al*, 2011), interacts with SUV39H1 (Aagaard *et al*, 1999; Yamamoto & Sonoda, 2003), SUV4-20H2 (Schotta *et al*, 2004; Souza *et al*, 2009), the DNA methylase DNMT1 (Fuks *et al*, 2003; Lehnertz *et al*, 2003; Smallwood *et al*,

1  Deutsches Krebsforschungszentrum (DKFZ) and BioQuant, Research Group Genome Organization & Function, Heidelberg, Germany
2  Deutsches Krebsforschungszentrum (DKFZ) and BioQuant, Division Theoretical Systems Biology, Heidelberg, Germany
3  Munich Center for Integrated Protein Science and Adolf Butenandt Institute, Ludwig Maximilians University, Munich, Germany
   *Corresponding author. Tel: +49 6221 5451376; Fax: +49 6221 5451487; E-mail: karsten.rippe@dkfz.de
   †Present address: Biochemistry & Structural Biology, Lund University, Lund, Sweden

2007) as well as the methyl-CpG-binding proteins MBD1 and MECP2 (Fujita *et al*, 2003; Agarwal *et al*, 2007). SUV39H1 interacts with the DNA methylation-associated proteins DNMT1, MBD1, and MECP2 (Lunyak *et al*, 2002; Fujita *et al*, 2003; Fuks *et al*, 2003; Esteve *et al*, 2006). Thus, a complex protein–protein interaction network exists in PCH. The interactions constituting this network in mammalian cells have been studied mostly *in vitro* or via immunoprecipitation experiments and have not been probed comprehensively in living cells.

Since HP1 interacts with SUV39H via its CSD, a feedback loop of HP1 binding-mediated H3K9 methylation has been proposed as a mechanism for propagating the H3K9me3 mark to adjacent nucleosomes (Schotta *et al*, 2002; Grewal & Jia, 2007; Eissenberg & Reuter, 2009). Theoretical models based on a combination of such feedback loops have suggested the existence of two discrete chromatin states that can stably co-exist ("bistability") for a certain range of conditions (Schreiber & Bernstein, 2002; Dodd *et al*, 2007; Angel *et al*, 2011). Hathaway *et al* have proposed an alternative, "monostable" model of heterochromatin propagation through interactions between neighboring nucleosomes (Hathaway *et al*, 2012). However, direct evidence on how such epigenetic networks might operate in living cells is lacking. In particular, three crucial questions remained unanswered: (i) How is the separation of the genome in active and silenced chromatin states established and maintained and what are the factors that provide *specificity* for distinct euchromatic and heterochromatic states? (ii) How is the *confinement* of a given chromatin state to a certain genomic locus achieved? For the case of a feedback loop between SUV39H, HP1, and H3K9me3 in PCH, it is elusive why the H3K9me3 does not spread throughout the whole genome. (iii) How is a given chromatin state like that of PCH transmitted through the cell cycle?

Here, we have set out to address these issues by dissecting the mouse pericentric heterochromatin network centered around the H3K9 and H4K20 methylation. This model system has the advantage that the corresponding heterochromatin domains can be readily identified on fluorescence microscopy images as chromatin-dense spots, the chromocenters. Accordingly, we were able to distinguish the features of PCH from the surrounding *bona fide* euchromatin. By applying a combination of fluorescence microscopy-based imaging, bleaching and correlation methods (Müller *et al*, 2009; Erdel *et al*, 2011b) in conjunction with quantitative mechanistic modeling, we identified distinct complexes of stably bound SUV39H

as the component that defines the PCH state. We further demonstrate that these SUV39H complexes represent "nucleation sites" that are sufficient to provide specificity, confined propagation of the H3K9me3 mark as well as cellular memory to transmit the PCH state through the cell cycle.

# Results

## The repressive PCH state is defined by enrichment of 5meC, MECP2, MBD1, and SUV39H

We quantitated the enrichment of the PCH-associated histone modifications H3K9me3, H4K20me3, the DNA methylation 5meC and the known proteins that set, remove, or recognize these modifications in the context of PCH in mouse NIH-3T3 fibroblasts: the histone H3K9 and histone H4K20 specific methylases SUV39H1, SUV39H2, SUV4-20H1, SUV4-20H2, all three isoforms of the H3K9me3-reader HP1 (HP1α, HP1β, and HP1γ), the histone demethylases JMJD2B and JMJD2C that remove H3K9me3, the DNA methylase DNMT1 as well as the methyl-CpG binding domain (MBD)-containing proteins MECP2, MBD1, MBD2, and MBD3 [see (Fodor *et al*, 2010) for a review of previously identified mammalian PCH components]. Furthermore, we included the transcription factors PAX3, PAX5, PAX7, and PAX9 in our analysis since a role for PAX3 in PCH assembly has been reported (Bulut-Karslioglu *et al*, 2012). Although the colocalization of these factors and histone marks with PCH was studied previously, a comprehensive quantitative analysis of their PCH-specificity and abundance has been lacking. Thus, we fluorescently labeled the factors of interest (Supplementary Fig S1) and determined the enrichment of GFP-tagged proteins in PCH using the workflow shown in Fig 1A: For each factor, the fluorescence intensity was measured in PCH as defined by foci with intense DAPI (4′,6-diamidino-2-phenylindole) staining and in the surrounding euchromatin of G1 phase cells. The enrichment of these factors in PCH was calculated, followed by normalization to the enrichment of core histone H2A and DAPI in PCH. Chromatin binding states were identified by fluorescence recovery after photobleaching (FRAP) and their enrichment in PCH was determined. Fluorescence correlation spectroscopy (FCS) was used to measure protein concentrations and the free diffusion coefficient in the cytoplasm, which served as a reference value. Protein enrichments, concentrations, chromatin

**Figure 1.  Quantitative analysis of core components of the pericentric heterochromatin (PCH) network.**

A   Workflow of integrative PCH network analysis. The enrichment of different factors in PCH was measured based on confocal laser scanning microscopy (CLSM) images. Abundance and chromatin interactions were determined by fluorescence fluctuation microscopy methods (FRAP, FCS). Additional information on protein–protein interactions was obtained as described in the text and in Fig 2. Based on the experimental data, a network model was developed that explains the stable maintenance of PCH.

B   Enrichment of proteins, DNA methylation, H3K9me3, and H4K20me3 in PCH over euchromatic regions from fluorescence intensity measurements in G1 phase cells. The red line marks the chromatin compaction in PCH (1.8-fold enrichment of H2A-RFP). Labeled H2A, SUV39H1, and HP1 isoforms were stably expressed, the other proteins were transiently expressed, and modifications were immunostained. Error bars correspond to standard error of the mean (SEM).

C   Enrichment of transiently bound and immobilized protein fractions in PCH versus euchromatin determined by FRAP. Red lines indicate 2-fold and 10-fold enrichment levels. Error bars correspond to standard deviation (SD).

D   Absolute concentrations of stably bound proteins in the immobile FRAP fraction.

E   FRAP experiments of transiently expressed MBD1 and MECP2 in PCH and euchromatin (Eu). Data were fitted to yield chromatin binding parameters.

F   FRAP of SUV39H1 (stably expressed) and SUV39H2 (transiently expressed).

G   FRAP of transiently expressed SUV4-20H1 and SUV4-20H2.

H   FRAP of stably expressed HP1β and HP1γ. For additional data including HP1α, see Supplementary Table S2.

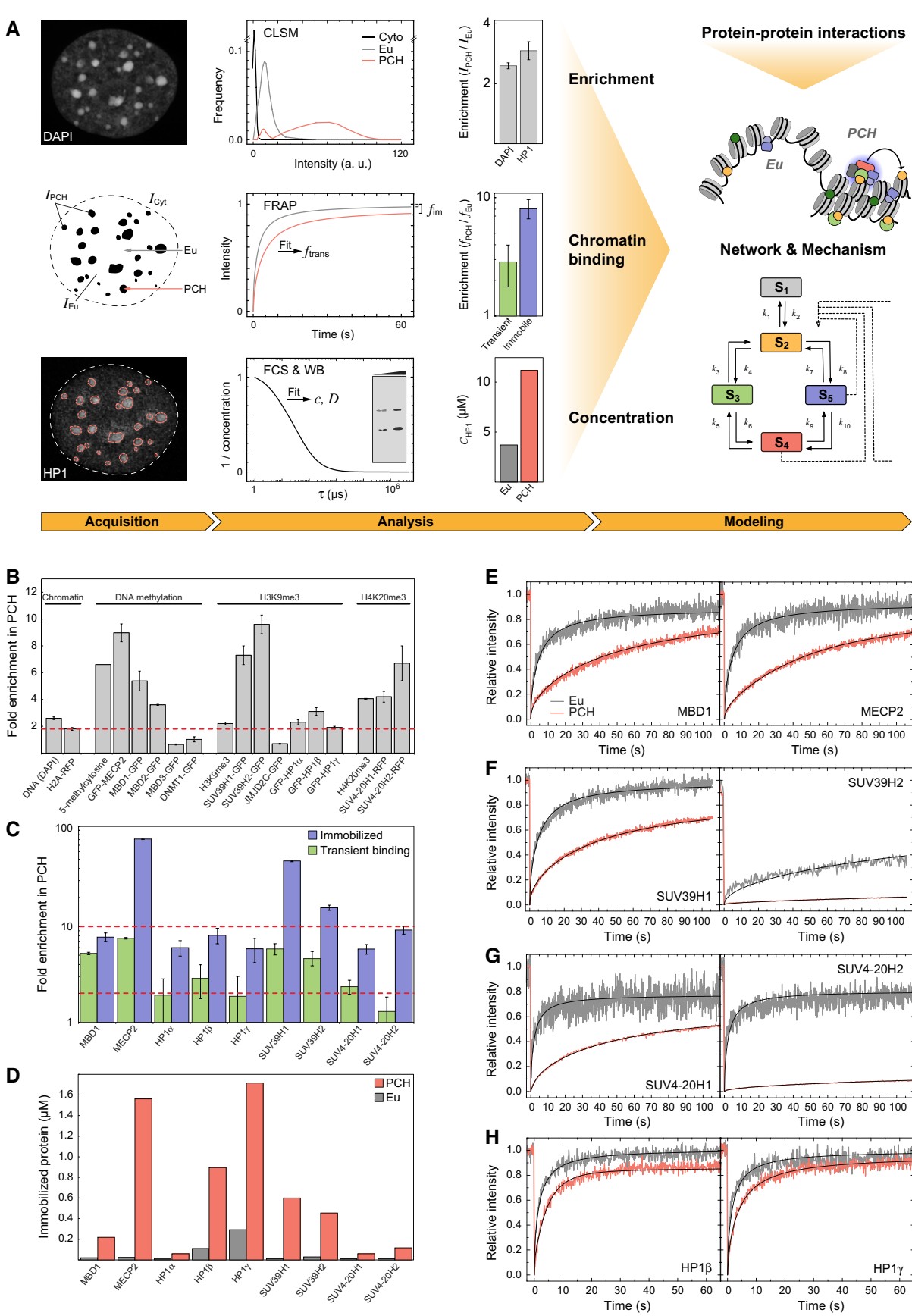

**Figure 1.**

binding states, and protein–protein interactions were integrated into a quantitative molecular model for PCH.

PCH was about $1.8 \pm 0.3$-fold denser than euchromatin as determined from the H2A-RFP intensity, whereas the DNA stain with DAPI yielded a $2.7 \pm 0.1$-fold enrichment in PCH, possibly reflecting its binding preference for A/T-rich sequences. We conclude that chromatin is about 2-fold more compacted in PCH (Fig 1B). In terms of factors targeted to PCH during G1, we identified three different groups (Fig 1B): (i) Proteins displaying a diffuse distribution throughout the nucleus or a slight depletion in PCH, which is indicative of little PCH-specific interactions (JMJD2B/C, DNMT1, MBD3, and the PAX proteins). (ii) Factors whose PCH enrichment (2–3-fold) essentially followed the increased chromatin density and accordingly displayed only moderate specificity for PCH (MBD2, HP1, and H3K9me3). When normalized to the H2A-RFP chromatin density, the average enrichment of H3K9me3 in PCH was only $1.4 \pm 0.2$-fold. Measured values ranged from 0.8- to 1.6-fold between different experiments and antibodies used (Millipore and Abcam). (iii) Factors that were clearly enriched above the 2-fold DNA compaction in PCH and thus represent PCH-specific components (5meC, MECP2, MBD1, SUV39H, H4K20me3, SUV4-20H: ~4- to 10-fold). The PCH enrichment of 5meC measured by immunostaining varied between 2- and 5-fold, depending on antibody and fixation protocols used. From bisulfite sequencing data of major satellite repeats in NIH-3T3 cells, mouse embryonic stem cells (ESCs), and primary differentiated cells, we calculated an enrichment of $7 \pm 1$ (Wilson & Jones, 1984; Yamagata *et al*, 2007; Meissner *et al*, 2008; Arand *et al*, 2012).

To validate the functional role of SUV39H and H3K9me3 in PCH of our cellular system, we measured the abundance of satellite transcripts by quantitative real-time PCR in ESCs, immortalized mouse embryonic fibroblasts (iMEFs) and iMEFs that had *Suv39h1* and *Suv39h2* (iMEF *Suv39h* dn) deleted (Peters *et al*, 2001). The transcription levels of pericentric major satellites were $4.6 \pm 0.2$-fold higher in ESCs and $14 \pm 2$-fold higher in iMEF *Suv39h* dn cells as compared to fully differentiated iMEF wild-type (wt) cells (Supplementary Fig S2A). Thus, PCH-specific H3K9me3 levels in wild-type iMEFs, ESCs, and *Suv39h* dn iMEFs anti-correlate with satellite repeat transcription, in agreement with previous measurements (Lehnertz *et al*, 2003; Martens *et al*, 2005; Meshorer & Misteli, 2006). Importantly, the chromatin-dense chromocenters persisted in both *Suv39h* and *Suv4-20h* double null cells, showing that transcriptional silencing is not due to chromatin compaction *per se* (Supplementary Fig S2B and C). Decondensation of the chromocenters was only observed upon inhibition of histone deacetylation (Supplementary Fig S2D) in agreement with previous reports (Taddei *et al*, 2001). In summary, we conclude that the enrichment of 5meC as well as MECP2, MBD1, SUV39H1, and SUV39H2 proteins represent the hallmarks of PCH.

## MECP2, SUV39H, and HP1 are the most abundant stably PCH-associated proteins

The above quantitative analysis of relative steady-state PCH enrichment levels lacks information on the absolute endogenous protein concentrations and does not resolve differences in binding kinetics. To address these issues, we integrated FCS and FRAP (Supplementary Fig S3A) (Müller *et al*, 2009). By combining quantitative FCS

and Western blot analysis, endogenous SUV39H and SUV4-20H concentrations were determined to be between 0.1–0.4 μM in euchromatin and 0.2–3.0 μM in PCH (Supplementary Fig S3B and C, Table 1, Supplementary Table S1). HP1 displayed the highest concentration of the factors studied of $19 \pm 12$ μM in euchromatin and $41 \pm 25$ μM in PCH, with HP1β and HP1γ being significantly more abundant than HP1α (Table 1, Supplementary Table S1). The diffusion coefficients measured by FCS in the cytoplasm were in the expected range and represent a reference value for the intracellular protein mobility of a given factor in the absence of chromatin interactions (Supplementary Table S1).

Chromatin binding in the nucleus was evaluated by FRAP (Fig 1E–H) with a reaction-diffusion analysis that yielded the effective diffusion coefficient $D_{eff}$ (including transient binding interactions), the dissociation rate constant $k_{off}$, the pseudo on-rate $k^*_{on}$ (including the free binding site concentration), and the protein fraction immobilized on the minute time scale (Supplementary Fig S3A). From these data, we calculated average protein residence times $\tau_{res}$ to different types of binding sites according to the following rationale (Table 1, Supplementary Tables S1 and S2): (i) The difference between $D_{free}$ and $D_{eff}$ indicated the presence of a protein pool, which binds transiently with $\tau_{res} \leq 0.5$ s. These represent the lowest affinity binding sites (class I) in our analysis. (ii) Kinetic on- and off-rates determined from the reaction-diffusion fit were used to characterize two additional types of binding sites, class II and class III, with $1/k_{off} = \tau_{res}$ in the range of 3–5 s (class II) and 25–100 s (class III). (iii) The highest affinity class IV binding sites comprised the protein fraction that was immobile during the measurement corresponding to a lower limit of $\tau_{res}$ of approximately 4 min.

Accordingly, we interpret transient interactions (class I) as unspecific chromatin binding interactions. These were present both in PCH and euchromatin and comprised essentially the entire protein pool of the H3K9me3 demethylases JMJD2B/C and the transcription factor PAX3 (Supplementary Table S2, Supplementary Fig S4A and B). Class II and class III binding sites were present in both euchromatin and PCH for the other proteins, albeit at different concentrations. With respect to the immobile protein fractions (class IV), we measured a particularly strong enrichment for MECP2 (~80-fold), SUV39H1 (~50-fold), and SUV39H2 (16-fold) in PCH as compared to euchromatin (Fig 1C). Immobile fractions of MBD1, SUV4-20H, and HP1 were about 8-fold enriched in PCH. When calculating the immobile fractions in terms of the absolute protein concentrations, the immobile fraction of the combined HP1 species was the largest (2.7 μM), followed by that of MECP2 (1.6 μM), SUV39H (1.1 μM), SUV4-20H (0.2 μM), and MBD1 (0.2 μM) (Fig 1D). Thus, the amount of immobilized HP1 provides enough protein molecules for interactions with MECP2/MBD1, SUV39H, and SUV4-20H at the high-affinity binding sites, although this HP1 fraction represents only approximately 7% of the total HP1 pool. Accordingly, we conclude that 1–3 μM of MECP2, SUV39H, HP1, and SUV4-20H are tightly bound ($k_{off} < 0.005$ s$^{-1}$) at PCH-specific sites that are mostly absent in euchromatin.

## PCH proteins form a complex interaction network

Our finding that MECP2/MBD1, HP1, and SUV39H are present at similar concentrations of about 1–3 μM in a stably PCH-attached

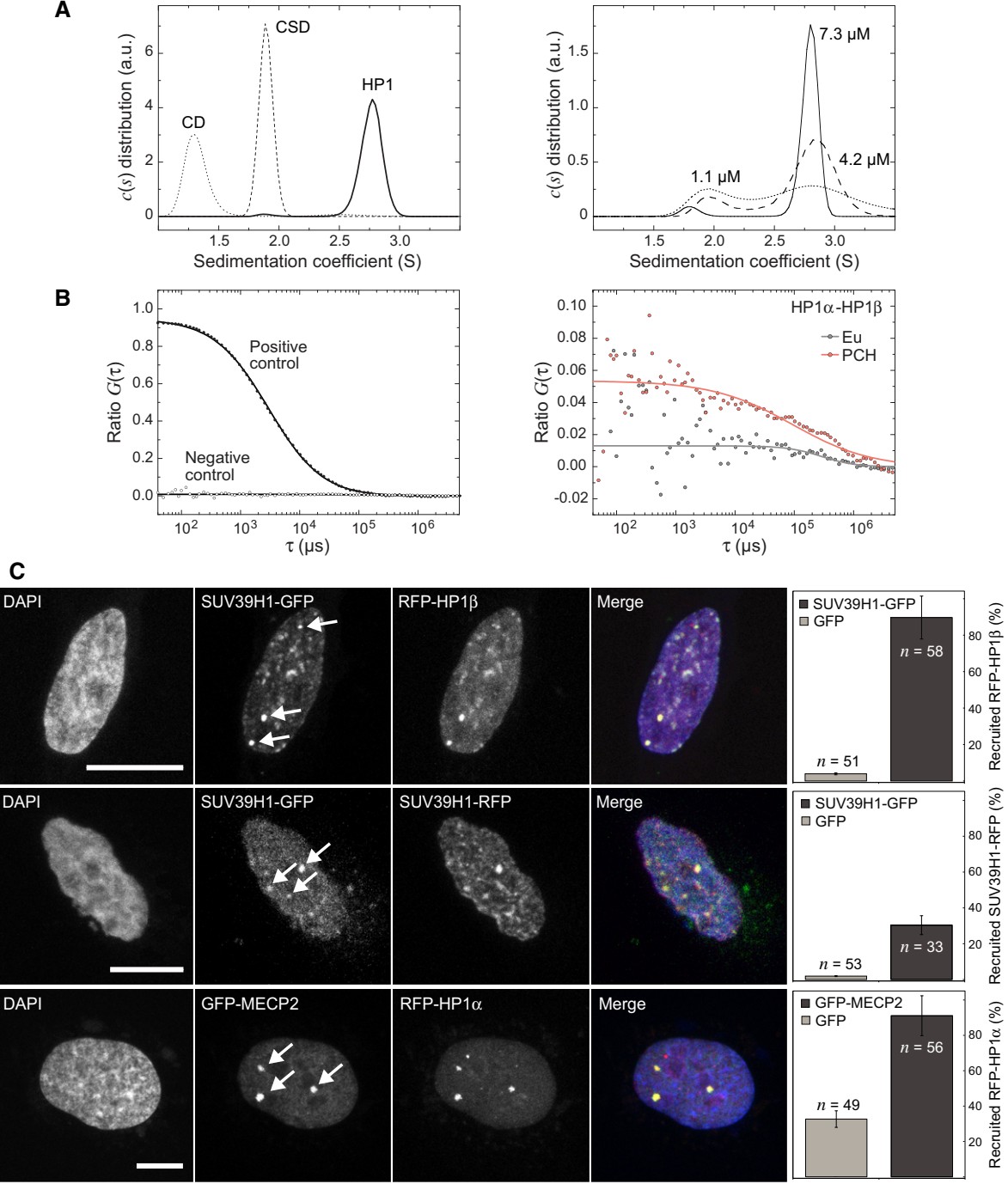

**Figure 2. Protein–protein interaction analysis of HP1 and SUV39H1.**

A  Analytical ultracentrifugation (AUC) experiments of HP1β and its isolated domains. The sedimentation coefficient *c*(s) distribution obtained from AUC sedimentation velocity runs in the concentration range from 8–30 μM showed dimerization of HP1β and the CSD (left panel) (Supplementary Table S3). Larger complexes were not observed. HP1β dimers dissociated at lower concentrations, which is reflected by two peaks in the *c*(s) distribution (right panel). An equilibrium dissociation constant of 1–2 μM was determined from the relative molar fractions of monomer and dimer species.

B  Protein–protein interaction analysis of soluble nucleoplasmic complexes by FCCS in transiently transfected NIH-3T3 cells. The parameter *ratio G*(τ) reflects the amount of complexes containing GFP- and RFP-labeled proteins. Control measurements were conducted with double-labeled beads in buffer (positive control) and with inert GFP and RFP transiently expressed in NIH-3T3 cells (negative control). For HP1, homodimers and heterodimers were found. Additional FCCS measurements for HP1 and SUV39H1 are shown in Supplementary Fig S5A.

C  F2H interaction analysis of SUV39H1 and other PCH proteins. Human U2OS cells were co-transfected with GBP-LacI and the indicated GFP and RFP constructs, resulting in tethering of the GFP-tagged protein to the three lac-operator integration sites. HP1β interacted with SUV39H1 in living cells with the percentage of colocalizations displayed in the barplot. Self-association of SUV39H1 and HP1α recruitment by MECP2 was also demonstrated. Isolated GFP was used as a negative control. Scale bars, 10 μm. Error bars correspond to SD. Further F2H interaction measurements are shown in Supplementary Fig S5B.

**Table 1.  Summary of binding interactions within the nucleus**

| | $C_{Eu}$ (µM) | $C_{PCH}$ (µM) | $C_{Cyto}$ (µM) | Binding classes | Binding Eu class, fract. (µM) | Binding PCH class, fract. (µM) | Class I[e] $\tau_{res}$ | Class II $\tau_{res}$ | Class III $\tau_{res}$ | Class IV[f] $\tau_{res}$ |
|---|---|---|---|---|---|---|---|---|---|---|
| MECP2 | 1.22[a] | 8.72[a] | 0.15[a] | 3 | class I: 75% (0.9) <br> class III[c]: 25% (0.3) | class I: 17% (1.5) <br> class III: 65% (5.7) <br> class IV: 18% (1.6) | ≤ 0.5 s | – | 25 s | ~4 min |
| HP1α | 0.5 | 1.0 | 0.06 | 3 | class I: 40% (0.2) <br> class II[e]: 60% (0.3) | class I: 24% (0.24) <br> class II: 70% (0.70) <br> class IV: 6% (0.06) | | | | |
| HP1β | 3.7 | 11.2 | 0.14 | 3 | class I: 40% (1.5) <br> class II[d]: 60% (2.2) | class I: 19% (2.1) <br> class II: 73% (8.2) <br> class IV: 8% (0.9) | ≤ 0.5 s | 3 s | – | ~4 min |
| HP1γ | 14.7 | 28.6 | 0.27 | 3 | class I: 99% (14.5) <br> class II[d]: 1% (0.15) | class I: 20% (5.7) <br> class II: 74% (21.2) <br> class IV: 6% (1.7) | | | | |
| SUV39H1 | 0.42 | 3.0 | 0.053 | 4 | class I: 32% (0.13) <br> class II: 65% (0.27) <br> class IV: 3% (0.01) | class I: 18% (0.54) <br> class II: 33% (0.99) <br> class III: 29% (0.87) <br> class IV: 20% (0.60) | ≤ 0.5 s | 5 s | 29 s | ~4 min |
| SUV39H2 | 0.12[a] | 0.88[a] | 0.016[a] | 2 | class I: 24% (0.03) <br> class IV: 76% (0.09) | class I: 9% (0.08) <br> class IV: 91% (0.8) | ≤ 0.5 s | – | – | ~4 min |
| SUV4-20H1 | 0.07[b] | 0.20[b] | 0.026[b] | 3 | class I: 85% (0.06) <br> class IV: 15% (0.01) | class I: 15% (0.03) <br> class III: 55% (0.11) <br> class IV: 30% (0.06) | ≤ 0.5 s | – | 25 s | ~4 min |
| SUV4-20H2 | 0.08[b] | 0.20[b] | 0.026[b] | 3 | class I: 83% (0.066) <br> class IV: 17% (0.014) | class I: 8% (0.016) <br> class III: 33% (0.066) <br> class IV: 59% (0.118) | ≤ 0.5 s | – | 100 s | ~4 min |
| JMJD2C | n.d. | n.d. | n.d. | 1 | class I | class I | ≤ 0.5 s | – | – | – |

Endogenous protein concentrations are given in terms of monomers. Values were determined from FCS measurements of GFP-tagged proteins (Supplementary Table S1) and APD-imaging (Supplementary Materials and Methods). The ratio of endogenous to exogenous protein concentrations in the cytoplasm (Cyto), euchromatin (Eu), and heterochromatin (PCH) was determined by quantitative Western blot analysis. The minimal number of binding classes was determined from FRAP measurements and subsequent data fitting to either a diffusion or a reaction-diffusion model. From the fit, fractions of each binding class were determined and residence times were calculated according to $\tau_{res} = 1/k_{off}$.

[a]Values for endogenous SUV39H2 and MECP2 concentrations were determined from RNA expression levels measured in iMEF cells relative to SUV39H1 levels.

[b]For SUV4-20H, concentrations were measured in embryonic stem cells and represent also values in fibroblasts since concentrations do not change significantly during differentiation (Efroni *et al*, 2008).

[c]Class III binding of MECP2 in euchromatin was estimated by refitting the FRAP curves with a reaction-diffusion model.

[d]Class II binding of HP1 was estimated by refitting the FRAP curves with a reaction-diffusion model including a fixed off-rate, which has been determined for heterochromatin.

[e]For unspecific binding of class I, the residence time $\tau_{res} \leq 0.5$ s is given as an estimate of the upper boundary resulting from the time resolution of the FRAP measurements (not extractable from data fitting).

[f]The immobile fraction (class IV) is measured from the plateau value of the FRAP curve after measurements for 4–5 min. It gives the lower boundary of the proteins' residence time for which an approximate value of 4 min was used in the network model.

state (Fig 1D; Table 1) suggests that they assemble into a complex. Accordingly, we mapped their protein interactions *in vitro* and in living cells (Fig 2, Supplementary Fig S5, Supplementary Tables S3 and S4): First, we used analytical ultracentrifugation (AUC) to measure the association state of full-length HP1β and its isolated CD and CSD at physiological ionic strength. Full-length HP1β formed a dimer with an equilibrium dissociation constant of 1–2 µM (Fig 2A, Supplementary Table S3). No larger complexes were detectable up to a concentration of 30 µM. Dimerization was mediated by the CSD of HP1 since the isolated domain was found to be dimeric while the CD was monomeric in agreement with previous results (Ball *et al*, 1997; Nielsen *et al*, 2001). Second, HP1 association in living cells was studied by fluorescence cross correlation spectroscopy (FCCS) after transfecting cells stably expressing GFP-HP1α with RFP-HP1α, β or γ. A soluble nuclear HP1 fraction of 65 ± 34% formed a dimer with either the same HP1 isoform (homodimer) or another isoform

(heterodimer) (Fig 2B, Supplementary Fig S5A). Self-association of SUV39H1 was shown with a corresponding approach, yielding an approximately 24% fraction of soluble SUV39H1 present in homo-dimeric complexes (Supplementary Fig S5A). Third, we applied the fluorescent two-hybrid (F2H) method (Zolghadr *et al*, 2008; Chung *et al*, 2011) to evaluate the SUV39H1-SUV39H1 self-association and interactions of HP1-dimers with stably chromatin-bound SUV39H in living cells. Tethering of SUV39H1-GFP to the *lac*O arrays resulted in the recruitment of RFP-HP1 and SUV39H1-RFP (Fig 2C, Supplementary Fig S5B). Furthermore, a strong interaction of HP1 with MECP2 and MBD1 was observed. However, we could not confirm association of SUV39H1 with the MBD-proteins that was reported elsewhere (Supplementary Fig S5B) (Lunyak *et al*, 2002; Fujita *et al*, 2003; Fuks *et al*, 2003; Esteve *et al*, 2006). A summary of all (direct or indirect) protein–protein associations for the factors studied here is given in Supplementary Table S4. It reveals that the

proteins involved in DNA and histone methylation form a complex interaction network in living fibroblasts. We conclude that several of the above described protein–protein interactions cooperate to target SUV39H to PCH and provide the interaction energy for its stable tethering.

### SUV4-20H operates downstream of SUV39H and stabilizes HP1 binding to PCH

SUV39H is required for the enrichment of H3K9me3, H4K20me3, SUV4-20H, and HP1 in PCH, and the loss of H3K9me3 and SUV39H leads to a strong increase in mobility of HP1 in PCH (Supplementary Figs S2B and S4C, Supplementary Table S5) as reported previously (Peters *et al*, 2001; Schotta *et al*, 2004; Müller *et al*, 2009). Immunostaining of 5meC revealed that this mark was maintained in iMEF *Suv39h* dn cells, and also the MBD-proteins MECP2 [as shown previously in (Brero *et al*, 2005)] and MBD1 remained enriched in PCH (Supplementary Fig S2B). This confirms that DNA methylation and enrichment of its reader proteins do not rely on SUV39H (Lehnertz *et al*, 2003; Brero *et al*, 2005).

To assess the influence of H4K20me3 on PCH, we analyzed heterochromatin proteins and histone modifications in iMEF *Suv4-20h* dn cells that lack both H4K20-specific methylases SUV4-20H1 and SUV4-20H2 (Supplementary Fig S2C) (Schotta *et al*, 2008). As expected, the H3K9me3 mark was unperturbed, and SUV39H as well as HP1 were still enriched at the chromocenters (Supplementary Fig S2C, Supplementary Table S5) (Schotta *et al*, 2004). However, FRAP analyses revealed that HP1 was more mobile in *Suv4-20h* dn cells as compared to wild-type cells (Supplementary Table S2). In particular, the immobile HP1 fraction was decreased to about half the wild-type value (Supplementary Fig S4D), the residence time of the bound fraction was shorter, and the effective diffusion coefficient increased. This suggests that SUV4-20H can enhance chromatin binding of HP1 although its binding to PCH occurs downstream of H3K9me3, SUV39H, and HP1.

### HP1 stabilizes SUV39H1 binding at PCH and promotes H3K9 trimethylation

We knocked down all three HP1 isoforms with siRNAs to evaluate the effect of HP1 on SUV39H binding and H3K9me3 levels in chromocenters versus euchromatin on the single-cell level. The knock-down resulted in a wide range of HP1 expression levels in individual cells (determined by immunostaining) that correlated with SUV39H1 and H3K9me3 intensity signals (Fig 3A). Global nuclear expression levels of stably integrated SUV39H1-GFP followed that of HP1, suggesting that HP1 stabilizes the *Suv39h1* transcript or protein. In addition, the PCH enrichment of SUV39H1 was reduced at lower HP1 concentrations. While SUV39H1 expression levels and PCH enrichment were sensitive to HP1 abundance over the complete range of knockdown concentrations, the H3K9me3 levels remained constant for HP1 levels above 80% of the wild-type concentration. Thus, cells were able to compensate for variations of HP1 and SUV39H1 concentrations to some extent. Below 80% of HP1, the heterochromatic H3K9me3 levels decayed gradually with decreasing HP1

expression until euchromatic levels were reached (Fig 3A and B). Thus, HP1 contributes to the enrichment of SUV39H1 in PCH and is required for maintaining wild-type H3K9me3 levels. Euchromatic H3K9me3 levels decreased slightly with reduced HP1 concentrations (Fig 3A). This could be related to interactions of HP1 with the histone methylases G9A and SETDB1 that are preferentially active in euchromatin (Tachibana *et al*, 2002; Chin *et al*, 2007; Loyola *et al*, 2009).

### SUV39H is responsible for depositing H3K9me3 preferentially in PCH

To quantify the contribution of SUV39H in catalyzing H3K9 trimethylation, we determined the H3K9me3 levels of euchromatin and PCH in wild-type and *Suv39h* dn iMEFs based on the relative H3K9me3 immunofluorescence signal in the same sample preparation (Fig 3C, Supplementary Fig S2B). In wild-type cells, an average H3K9me3 level of $38 \pm 3\%$ in PCH and $28 \pm 1\%$ in euchromatin was calculated based on a total H3K9me3 level of 28% reported previously for mouse fibroblasts (Fodor *et al*, 2006). The corresponding values in the *Suv39h* dn cells were $13 \pm 2\%$ (PCH) and $25 \pm 4\%$ (euchromatin), from which the relative methylation rate of SUV39H can be estimated according to a simple quantitative model: Nucleosomes can carry H3K9me3 (M) or lack this modification (U) (Fig 3D, Supplementary Materials and Methods). Since the JMJD2B/C demethylases were homogeneously distributed in the nucleus and displayed similar mobility in both euchromatin and PCH (Supplementary Fig S4A), the demethylation rate $k_{\text{-m}}$ is assumed to be equal in both chromatin states. The resulting H3K9me3 level in each state is solely determined by the ratio of methylation rate $k_{\text{m}}$ to $k_{\text{-m}}$. This yields an 8-fold preference of SUV39H for methylation of PCH versus euchromatin, while the euchromatin-specific methylation provided by other methylases like SETDB1 and G9A is roughly 2-fold higher in euchromatin than in PCH (Fig 3D). Notably, the SUV39H specificity for PCH correlates well with the enrichment of chromatin-bound SUV39H molecules in PCH (16- to 50-fold, Fig 1C). Thus, we conclude that the SUV39H-dependent H3K9me3 modification is directly related to the amount of chromatin-bound enzyme.

### Silenced major satellite repeats are enriched with SUV39H, H3K9me3, HP1, and 5meC

To corroborate that SUV39H binding to chromatin concurs with HP1 and H3K9me3 at sites of DNA methylation, we conducted an analysis by ChIP-seq (chromatin immunoprecipitation followed by DNA sequencing). The enrichment of HP1β, SUV39H1, SUV39H2, and H3K9me3 at major satellite repeats was evaluated in neural progenitor cells (NPCs) using the H3K36me3 modification as a reference signature for transcriptionally active chromatin. NPCs were generated *in vitro* from ESCs and represent a well-established model system for studying epigenetic modifications and chromatin composition established during differentiation (Teif *et al*, 2012; Lorthongpanich *et al*, 2013). SUV39H1, SUV39H2, H3K9me3, and HP1β were enriched at the canonical major satellite repeat sequence while no enrichment was detected for H3K36me3 (Fig 4A and B). Furthermore, we analyzed the distribution of bound

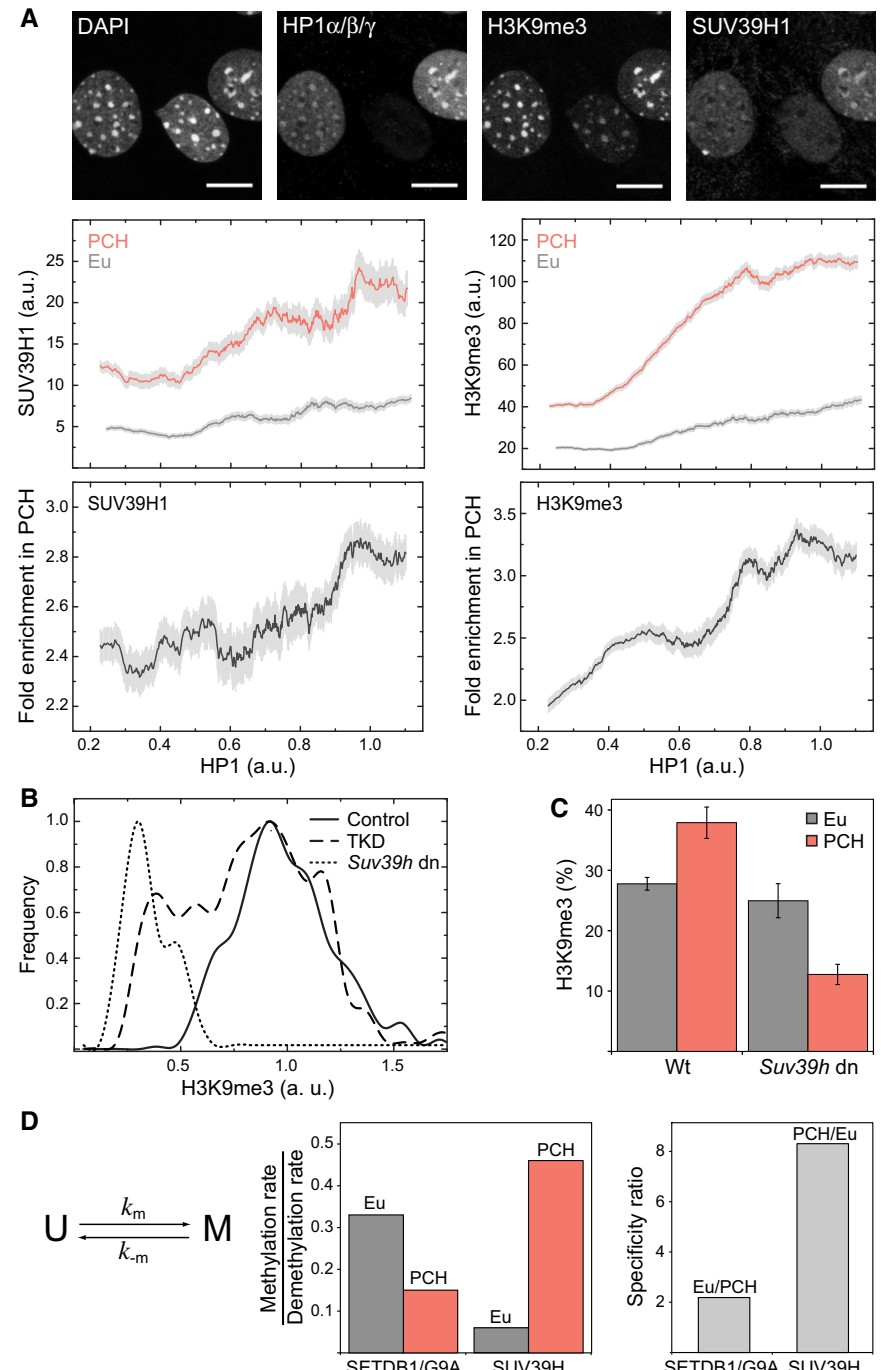

**Figure 3.  Effect of perturbation of HP1 and SUV39H protein expression.**

A   Triple knockdown of HP1α/β/γ by siRNA in NIH-3T3 cells stably expressing SUV39H1-GFP. As apparent from HP1 immunostaining (top) knockdown efficiencies varied largely between individual cells so that a large range of endogenous HP1 concentration was covered. The effects on the abundance of SUV39H1-GFP and the H3K9me3 mark were detected by immunostaining (upper plot panels). Their fold enrichment within PCH is presented in the bottom panels. Light gray bars depict SEM.

B   Frequency distribution of methylation levels per PCH focus in wild-type (wt) cells (control, solid line), *Suv39h* dn cells (*Suv39h* dn, dotted line), and HP1 triple-knockdown cells (TKD, dashed line) transfected with HP1 siRNA.

C   Comparison of H3K9me3 levels in wild-type and *Suv39h* dn cells determined from quantitative imaging after immunostaining. H3K9me3 in PCH was largely reduced in *Suv39h* dn cells while euchromatic levels were unperturbed (see also Supplementary Fig S2B). Absolute H3K9me3 levels were calculated based on the global cellular H3K9me3 level reported previously (Waterston *et al*, 2002; Fodor *et al*, 2006). Error bars correspond to SEM.

D   Average methylation rates were determined based on a simple model assuming an unmethylated and a methylated state for each nucleosome. From the steady-state levels of H3K9me3 in wild-type and *Suv39h* dn cells, the relative rate constants for the transitions are derived. SUV39H methylates PCH with a specificity ratio of 8, whereas SETDB1/G9A methylate preferentially euchromatic regions but with a lower specificity ranging from 1.5–2.2.

proteins at 16 uniquely mappable intergenic/intronic major satellite repeats annotated by the RepeatMasker tool (Fig 4C and D). Of these repeats, 12 were enriched for SUV39H, H3K9me3, and HP1β while H3K36me3 was depleted. Another two repeats resided in an inactive state with low H3K36me3 and higher H3K9me3 levels but lacked enrichment of SUV39H or HP1β. In contrast, the two active repeats carrying H3K36me3 lacked SUV39H, H3K9me3, and HP1β. Thus, the ChIP-seq analysis corroborates our conclusions that stable chromatin binding of SUV39H and transcriptional repression

of major repeats correlate with the presence of HP1 and H3K9me3. Based on the 5meC distribution reported elsewhere (Lorthongpanich *et al*, 2013), we calculated CpG methylation levels of $88 \pm 8\%$ at the 12 inactive intergenic/intronic repeats loaded with SUV39H, H3K9me3, and HP1. The two active repeats had similar CpG densities and similar CpG methylation levels of $86 \pm 2\%$. Since the normalized 5meC density was only slightly higher at $1.3 \pm 0.1$-fold for silenced repeats compared to transcribed ones, we conclude that 5meC is not the dominant silencing factor for these repeats.

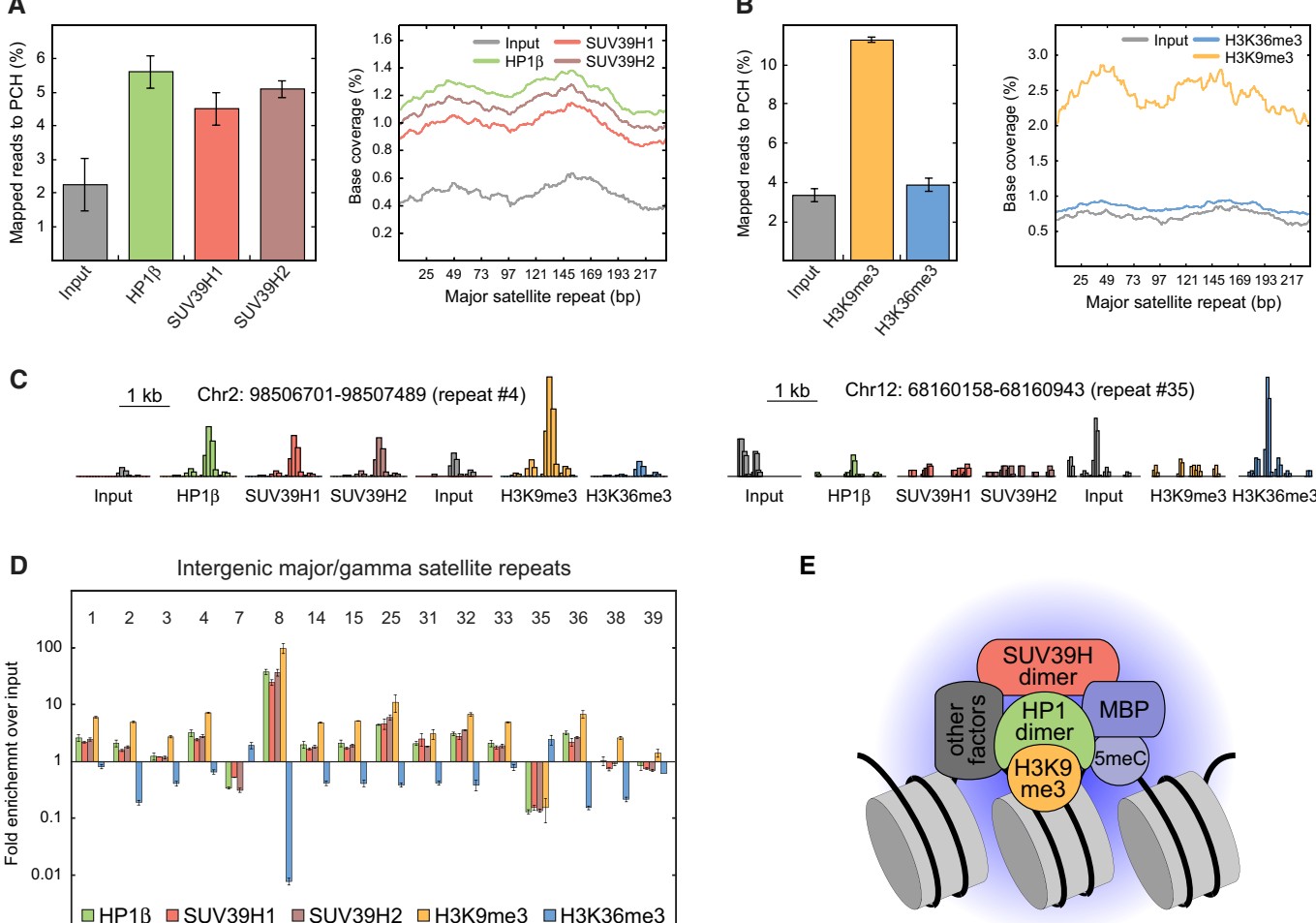

**Figure 4. Genome-wide ChIP-seq analysis of HP1, SUV39H, and H3K9me3 enrichment at major satellite repeats.**

A   Fraction of total sequencing reads that mapped to the consensus sequence of major satellite repeats (gamma satellites, GSAT) after ChIP-seq of HP1β, SUV39H1, or SUV39H2. Mild sonication conditions were applied in neural progenitor cells (NPCs). The histogram shows the distribution of the mapped reads along the consensus sequence. Error bars correspond to SD.

B   Same as panel A but for ChIP-seq of H3K9me3 and H3K36me3 and with stronger sonication conditions.

C   Representative sequencing read distributions at interspersed major satellites that can be uniquely identified in the genome. Two repeats located on chromosomes 2 and 12 (no. 4 and no. 35 in panel D) are depicted. HP1 and SUV39H were enriched at the transcriptionally inactive repeat (no. 4) that additionally had a high level of the repressive H3K9me3 mark and a low level of the activating H3K36 trimethylation. At the actively transcribed repeat (no. 35) marked by H3K36me3, binding of SUV39H and HP1 was at background levels. The height of the histogram is normalized to total reads.

D   Enrichment of all mapped sequencing reads to interspersed GSAT regions over input control. The 16 regions that can be uniquely mapped are shown. Error bars correspond to SD.

E   Composition of the HP1-SUV39H nucleation complex as inferred from the enrichment of stably bound protein in PCH, protein–protein interaction measurements and ChIP-seq analysis. The postulated complex comprises a HP1 dimer binding H3K9me3-modified nucleosomes and interacting with a SUV39H dimer. Binding of SUV39H is further stabilized by methyl-binding proteins (MBPs) that recognize 5meC and possibly by other interacting factors.

## SUV39H methylases remain attached to chromatin during mitosis

Many proteins remain attached to PCH during mitosis (Lewis *et al*, 1992; Fujita *et al*, 1999; Bachman *et al*, 2001; Craig *et al*, 2003; Hayakawa *et al*, 2003; Easwaran *et al*, 2004; Kourmouli *et al*, 2004; Mateescu *et al*, 2004; Brero *et al*, 2005; Fischle *et al*, 2005; Hirota *et al*, 2005; McManus *et al*, 2006; Schermelleh *et al*, 2007; Hahn *et al*, 2013). Thus, the PCH state might be transmitted through cell division by stably bound bookmarking factors. Since SUV39H1 also remains bound to chromosomes in metaphase (Melcher *et al*, 2000), we analyzed its localization in all mitotic phases (Fig 5A). Notably, SUV39H1 remained attached to chromatin through mitosis. From FRAP experiments on mitotic cells, we found $5 \pm 3\%$ of SUV39H1 stably bound to chromatin (Fig 5B, Supplementary Table S6). Since euchromatin and PCH cannot be distinguished on mitotic chromosomes, this should be compared to the weighted average of bound SUV39H1 in PCH and euchromatin of interphase cells ($4 \pm 1\%$). Thus, most SUV39H remained stably bound, suggesting that it serves a bookmarking function for PCH.

The cell cycle-dependent chromatin interactions of HP1 and SUV39H1 were further analyzed by FRAP (Supplementary Table S6). Co-transfection of RFP-tagged proliferating cell nuclear antigen (PCNA) served as a marker for G1 and S phase and delocalized HP1 as an indicator for G2 phase. SUV39H1 showed relatively similar chromatin interaction throughout interphase (Supplementary Fig S1C, Supplementary Table S6). All HP1 isoforms had comparable cell cycle-dependent mobility, and HP1α is shown as an example (Supplementary Table S6). While HP1 exhibited considerable chromatin binding in G1 and S phase consistent with previous findings (Cheutin *et al*, 2003; Festenstein *et al*, 2003; Schmiedeberg *et al*, 2004; Dialynas *et al*, 2007), HP1 mobility could best be described by a diffusion model for both euchromatin and PCH in G2, indicating significantly reduced chromatin interactions. The immobile fraction in PCH was reduced to euchromatin values of $3 \pm 2\%$. The remaining fraction of approximately 3% stably tethered HP1 corresponds to a concentration of 1.2 μM, which is similar to the amount of immobile SUV39H and SUV4-20H in PCH. In summary, our combined imaging- and FRAP-based analyses provide evidence that a considerable fraction of SUV39H remains stably bound to chromatin throughout the cell cycle including all phases of mitosis. Likewise, H3K9me3, HP1, 5meC, MECP2, and MBD1 remained available to enhance SUV39H chromatin interactions. This is consistent with an inheritance mechanism in which the PCH state is transmitted via chromatin-bound bookmarking factors.

## Stable SUV39H-containing complexes in PCH arise from multiple protein–protein interactions

Our protein–chromatin and protein–protein interaction data can be rationalized with a model in which SUV39H and HP1 can bind either separately to weaker affinity binding sites with $\Delta G_{SUV39H}$ and $\Delta G_{HP1}$, respectively, or to high-affinity sites where both proteins interact with chromatin and are additionally linked by protein–protein interactions. This interaction could be further stabilized by MECP2 or MBD1, which themselves display high-affinity PCH binding, or by other additional factors. The binding free energy $\Delta G_{HP1\text{-}SUV39H}$ of an HP1-SUV39H complex to these high-affinity sites can be approximated as $\Delta G_{HP1} + \Delta G_{SUV39H}$. Based on our FRAP data

(Supplementary Table S2), dissociation constants and residence times for the individual binding reactions to H3K9 trimethylated nucleosomes are $K_d = 19$ μM and $\tau_{res} = 3$ s for HP1, and $K_d = 4$ μM and $\tau_{res} = 13$ s for SUV39H1. Together, these contributions lead to a highly stable chromatin-bound HP1-SUV39H complex with $K_d \approx 0.07$ nM and a corresponding residence time of several minutes, which is consistent with the immobile fraction in our FRAP experiments. Based on the limiting concentration measured for immobilized SUV39H (Table 1, Supplementary Table S7), this complex can only be sparsely distributed throughout PCH, that is the stoichiometry equals one complex per 170 nucleosomes.

## Immobilized SUV39H1 is sufficient to form a *de novo* H3K9me3 domain

The correlation between the highly enriched stably chromatin-bound SUV39H fraction (Fig 1C) and the steady-state H3K9me3 levels (Fig 3C) in PCH prompted us to ask whether these stably bound complexes are responsible for the deposition of H3K9me3 in PCH. To test if immobilized SUV39H1 is sufficient to establish a methylated chromatin domain, GFP-SUV39H1 was tethered to the nuclear lamina via a GFP-binding protein that was fused to Lamin B1 (Rothbauer *et al*, 2008) in living iMEF *Suv39h* dn cells. Subsequently, the localization of H3K9me3 was monitored with an RFP-tagged chromodomain (CD) of HP1 (Fig 6A and B). A quantitative analysis of the averaged H3K9me3 profile across the lamina revealed *de novo* H3K9me3 modifications at the nuclear lamina for wild-type SUV39H1 but not for the inactive SUV39H1-H324L-mutant (Fig 6A and C). Thus, immobilized SUV39H methylates nucleosomes that are brought into spatial proximity. The slightly increased width of the profile at half maximum for H3K9me3 ($0.47 \pm 0.03$ μm) versus that of GFP-SUV39H1 ($0.40 \pm 0.02$ μm) indicated that the newly formed H3K9me3 regions extended for $< 0.1$ μm into the nuclear interior. Thus, the trimethylation mark did not spread beyond those chromatin loci that could transiently interact with tethered SUV39H1 via chromatin dynamics, although free GFP-SUV39H1 was present in the nucleoplasm as detected by FCS.

We conclude that the endogenous propagation of H3K9me3 in PCH can originate from relatively sparsely distributed immobilized SUV39H complexes that locally extend H3K9me3 to spatially adjacent sites. The underlying molecular spreading mechanism might be chromatin looping that can efficiently promote interactions within limited genomic distances of several kilobases or $< 100$ nm (Rippe, 2001; Erdel *et al*, 2013).

## The dynamics of protein binding and histone modifications in PCH can be integrated into a quantitative network model

Based on our experimental observations and data analysis, we developed a quantitative model for the epigenetic network centered around H3K9me3 in PCH with all parameters compiled in Supplementary Table S7. It is based on stably bound SUV39H complexes that are specifically tethered to PCH via multiple interactions (Fig 4E). These SUV39H complexes represent nucleation sites that mediate confined propagation of H3K9me3 via chromatin looping (Rippe, 2001; Erdel *et al*, 2013) (Fig 7A). In particular, such a mechanism rationalizes the experimental findings on the limited extension of H3K9me3 from

                    

**A**

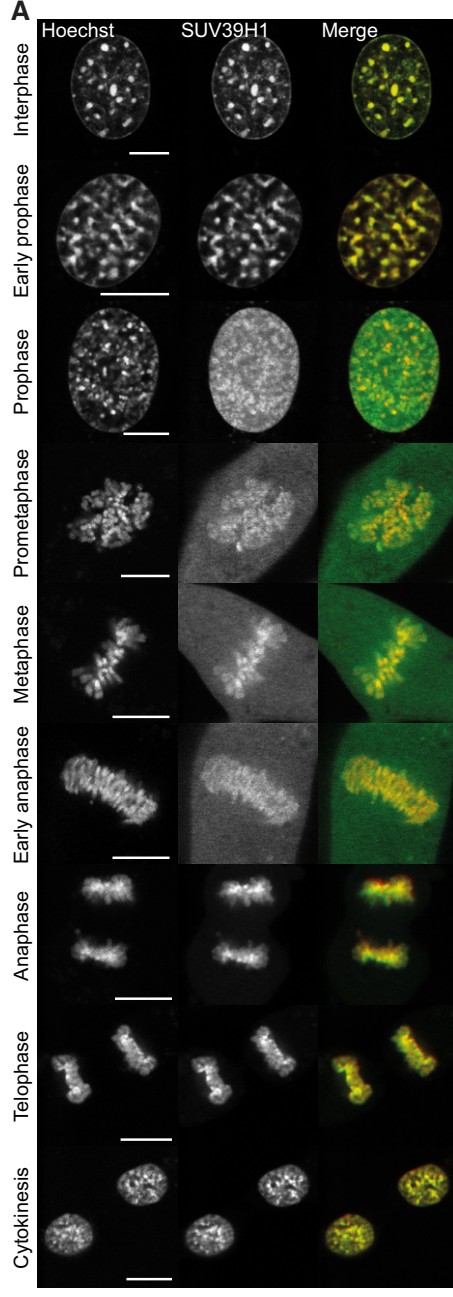

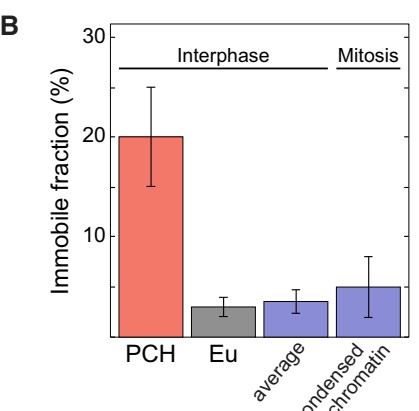

**Figure 5.   Chromatin bookmarking by SUV39H1 during the complete cell cycle.**

A   Nuclear distribution of SUV39H1-GFP in living cells during different cell cycle phases. SUV39H1-GFP remained stably associated to mitotic chromatin (stained with Hoechst 33342) during all phases of the cell cycle. Scale bars, 10 μm.

B   FRAP measurements of SUV39H1-GFP bound to condensed chromosomes in mitotic cells. Since euchromatin and PCH cannot be distinguished on mitotic chromosomes, the immobilized fraction of $5 \pm 3\%$ was compared to the weighted average of immobile SUV39H1 in PCH and euchromatin in interphase cells (Supplementary Tables S2 and S6). Error bars correspond to SEM.

SUV39H1 bound to the nuclear lamina (Fig 6) as well as the size of histone methylation domains that have been found around chromatin-tethered proteins (Erdel *et al*, 2013).

Our quantitative model focuses on the signature heterochromatin mark H3K9me3 and its interacting proteins, HP1 and SUV39H, in an extended chromatin segment (300 nucleosomes or ~60 kb DNA). Each nucleosome can either reside in the unmodified state "*n*" or in the H3K9 trimethylated state "*m*" (Fig 7B). Depending on the presence of HP1 ("*H*") and SUV39H ("*S*"), the chromatin-bound complexes "*Hm*", "*Sm*", and "*SHm*" can assemble. The PCH-specific high-affinity binding sites for SUV39H and HP1 are referred to as origins "*o*", which can either be free, occupied by SUV39H ("*So*") or HP1 ("*Ho*") alone, or by the HP1-SUV39H complex ("*SHo*"). According to our experimental data, every 8th nucleosome in PCH and every 71st nucleosome in euchromatin is an origin (see Supplementary Materials and Methods, Mathematical modeling of PCH network). Based on the experimentally measured immobile SUV39H fractions and concentrations, every 21st origin, that is every 170th nucleosome, is in the *SHo* state at a given point of time. All parameters used in the modeling and their sources are given in Supplementary Table S7.

The model quantitatively accounts for the different experimentally observed types of binding sites for SUV39H1 and HP1 and their occupancy. For the H3K9 trimethylation reaction, we distinguish different pathways. Freely mobile SUV39H at a concentration $c_s$ methylates nucleosomes with a basal rate $k_m \cdot c_s$. H3K9 trimethylation by other histone methylases like G9A/SETDB1 is accounted for as an additional reaction with rate constant $k_u$ in PCH and $k_e + k_u$ in euchromatin. Finally, chromatin-bound SUV39H catalyzes H3K9 trimethylation via chromatin looping. The efficiency for trimethylating H3K9 in a nucleosome at distance $b$ from the SUV39H-bound site corresponds to $k_m \cdot j_M(b)$. Here, $j_M(b)$ is the local concentration of the respective complex in proximity of the target nucleosome (Erdel *et al*, 2013) (Fig 7A). Importantly, we do not assume that chromatin-bound SUV39H is intrinsically more active than free SUV39H but that the decisive factor is the enhanced local concentration of the former.

For our model, the association and dissociation rates and the concentrations of free proteins were taken as determined from the FRAP and FCS experiments (Supplementary Materials and Methods, Supplementary Table S7). To simplify the model, we combined the parameters for SUV39H1 and SUV39H2 weighted according to their concentrations into a unified SUV39H enzyme (Supplementary Table S7). The rate constant for H3K9me3 demethylation was fixed to $k_{-m} = 0.0013$ min$^{-1}$ according to mass spectrometry experiments in HeLa cells (Zee *et al*, 2010). The unspecific methylation rate $k_u$, the

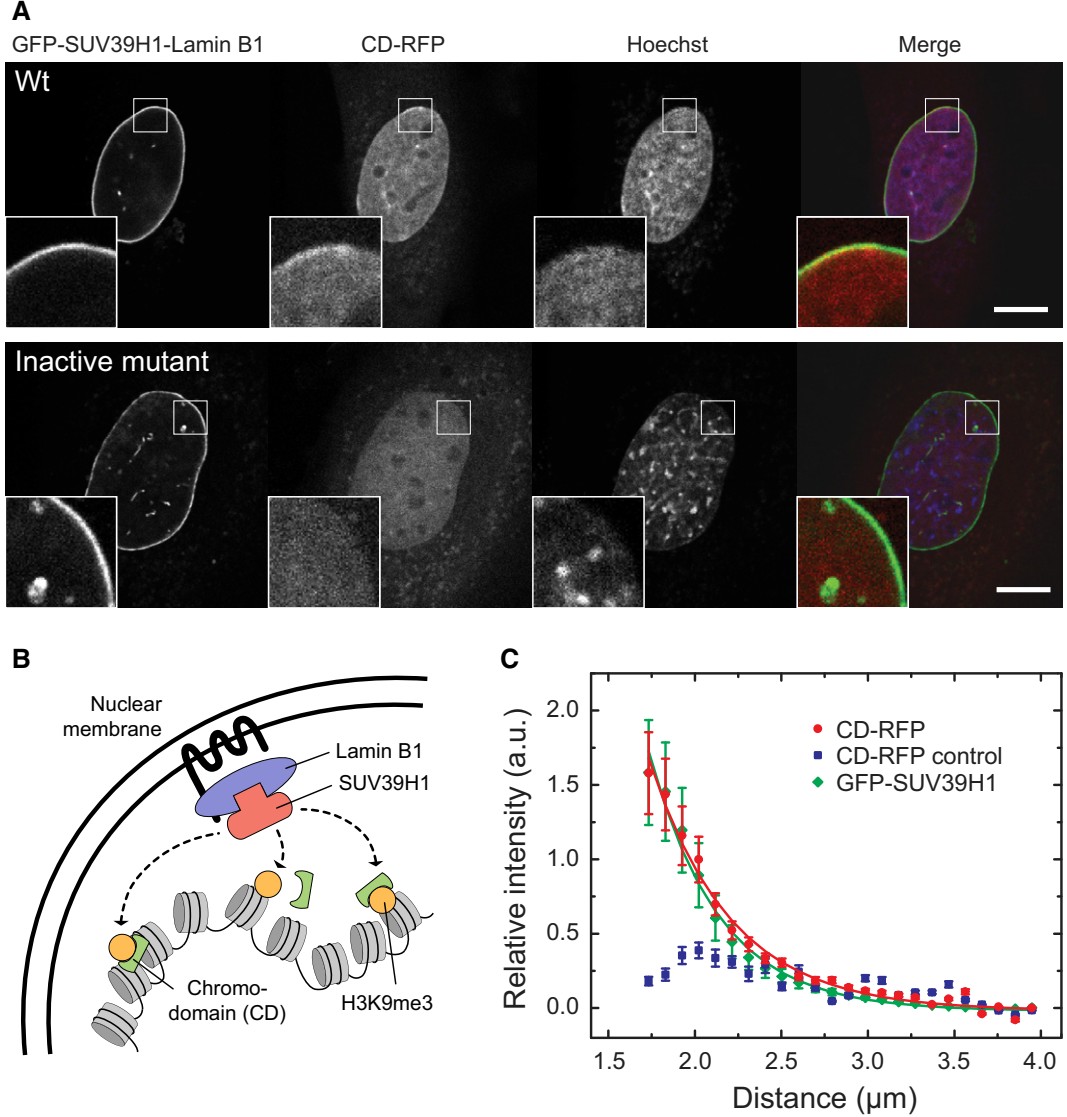

**Figure 6.  Propagation of H3K9me3 by stably tethered SUV39H1 to nucleosomes in its spatial proximity.**

A   Establishment of H3K9me3 domains by GFP-SUV39H1 recruited to the nuclear lamina via GBP-Lamin B1 in iMEF *Suv39h* dn cells. H3K9me3 was detected via CD-RFP and appeared in a confined region adjacent to the nuclear lamina (see inset). When the inactive mutant SUV39H1-H324L-GFP was recruited, no enrichment of CD-RFP was observed. Scale bar, 10 μm.

B   Cartoon model depicting the experimental setup in panel A.

C   Averaged radial fluorescence intensity profiles from the lamina to the center of the nucleus measured for the experiments described in panel A. The profile of CD-RFP reflects the H3K9me3 levels and was measured in cells transfected with GFP-SUV39H1 (red) or the inactive SUV39H1-H324L-GFP mutant (blue, control). The recruitment of GFP-SUV39H1 (green) resulted in a lamina-confined enrichment with a width of $0.40 \pm 0.02$ μm as determined by fitting the data to an exponential decay curve. While wild-type SUV39H1 methylated the surrounding chromatin within a confined area of $0.47 \pm 0.03$ μm width (red), SUV39H1-H324L (blue) did not increase the methylation (blue) in this region. Error bars correspond to SEM.

euchromatin-specific rate $k_e$, and the SUV39H-dependent methylation rate $k_m$ were fitted to yield the measured steady-state levels of H3K9me3 in PCH (38%) and euchromatin (28%) of wild-type cells (Fig 3C). The system was formulated as a set of deterministic ordinary differential equations (ODEs) to calculate the steady-state probabilities of each nucleosome to be in a particular state for a given set of conditions and, in parallel, simulated stochastically (Fig 7C and D; Supplementary Model Code). The model simulations show that the relatively sparse binding events of SUV39H at the origin sites

are sufficient to account for the increased H3K9me3 levels in PCH compared to euchromatin (Fig 7C and D). At any given nucleosome, the methylation state fluctuates over time (Fig 7D). However, the time average is consistently described by the stochastic and deterministic models and yields the observed enrichment of H3K9me3 in PCH over euchromatin as well as the measured occupancies of HP1 and SUV39H. Thus, we conclude that PCH-specific high-affinity SUV39H/HP1 binding sites can robustly maintain the PCH-specific H3K9me3 levels via a DNA looping mechanism.

    

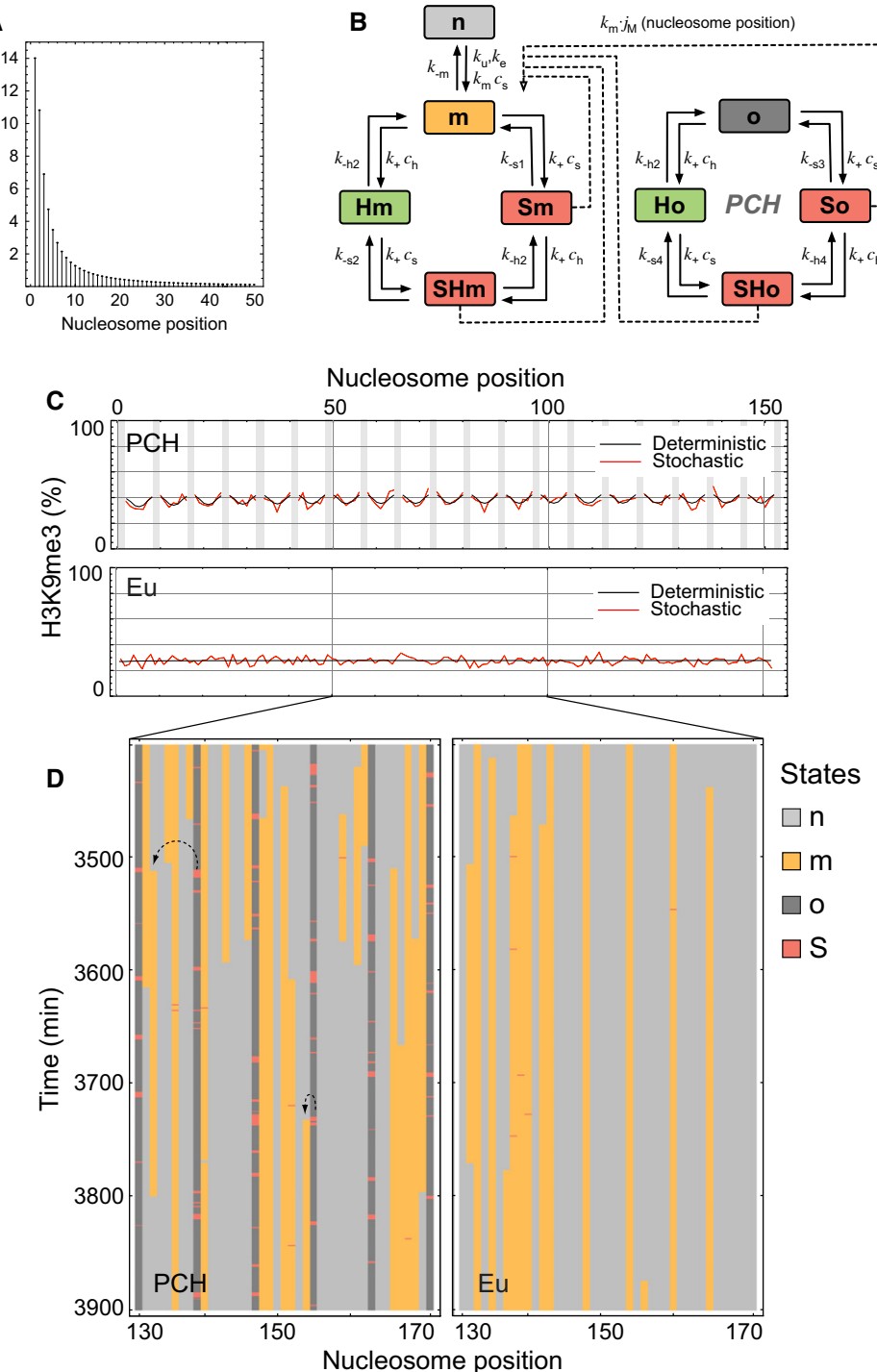

**Figure 7. Quantitative model for H3K9 trimethylation in PCH.**

A  Probability of interactions between two nucleosomes due to chromatin looping as expressed by their local molar concentration $j_M$. Based on the data for chromatin interactions during recombination in living cells (Ringrose *et al*, 1999), the dependence of $j_M$ on the separation distance from a SUV39H-bound nucleosome was calculated as described previously (Rippe, 2001; Erdel *et al*, 2013).

B  PCH network model scheme. The model includes transitions between the following states according to the indicated rate constants: "*n*", nucleosome without H3K9me3; "*m*", H3K9me3 modified; "*o*", origin that represents a high-affinity binding site for SUV39H. HP1 and SUV39H bound to these sites are represented by *H* and *S*, respectively, to form the states "*Hm*", "*Sm*", "*SHm*", "*Ho*", "*So*", and "*SHo*". All SUV39H-bound states (red) enhance methylation of adjacent unmodified nucleosomes through chromatin looping. See text and Supplementary Table S7 for further details.

C  Steady-state distribution of methylation levels for a chromatin segment in PCH or euchromatin. The deterministic and stochastic distributions are depicted in black and red, respectively. Stochastic solutions were averaged over 5,000 min. The nucleation origins (both occupied and unoccupied) are marked in gray.

D  Stochastic simulation for a region of 40 nucleosomes. Same color code for nucleosome states and origins as in panel B.

    

## Stochastic simulations reveal that PCH-specific features are robustly maintained for the experimentally determined parameter range

Several experimental and theoretical studies showed that the epigenetic silencing mechanisms can be sensitive to stochastic effects either due to the presence of positive feedback loops in nucleosome modification (Sneppen *et al*, 2008), long-range nucleosomal interactions (Dodd *et al*, 2007), the cooperativity in recruitment of histone modifiers (Sedighi & Sengupta, 2007) or simply due to low protein concentrations. Accordingly, we evaluated our model with respect to the degree of intrinsic noise that can cause stochastic focusing effects by conducting Monte Carlo simulations according to the Gillespie stochastic simulation algorithm. An example for the resulting fluctuations of the states in which individual nucleosomes are present in PCH or in euchromatin is depicted in Fig 7D. Relatively sparse binding events of SUV39H at the "*o*" sites are sufficient to increase H3K9me3 levels compared to euchromatin. For both the PCH and euchromatin state, we simulated 100 individual time traces up to 5,000 min starting from an initially naive non-modified fiber (Fig 7D). The low variability of H3K9me3 in steady state revealed that the level of intrinsic noise in our model is very low when averaged over the entire nucleosome chain, with a standard deviation of approximately 3% from the population-mean for PCH. As expected, the spatial steady-state distribution of H3K9me3 modifications over the entire chain agreed well with the deterministic solutions for all conditions (Fig 7C). Thus, we conclude that the system robustly maintains the PCH-specific H3K9me3 levels.

## The cellular response to perturbations of the H3K9me3 state is accurately predicted by the network model

We systematically investigated the predictions of our model with respect to the response of the system to perturbations induced by (i) lowering the HP1 concentration, (ii) increasing the H3K9me3 demethylation activity or (iii) changing the concentration of origin sites. First, we evaluated the distribution of H3K9me3 levels in model simulations for different HP1 concentrations. SUV39H binding and H3K9me3 levels gradually decreased when lowering the HP1 concentrations, consistent with the behavior observed in the HP1 knockdown experiments (Fig 8A and B). Second, we mimicked the increase of H3K9me3 demethylation activity by raising the corresponding parameter $k_{m}$. This resulted in a gradual decrease of H3K9me3 in PCH, which contradicts the notion of bistable states. To test this behavior of the system experimentally, we overexpressed JMJD2C, an H3K9me3-specific histone demethylase, to reduce H3K9 trimethylation (Fig 8C). H3K9me3 levels were evaluated in PCH of single cells on fluorescence microscopy images, normalized to the DAPI signal, and correlated with the JMJD2C-GFP expression levels. The relation of demethylation rate and H3K9me3 levels predicted from our model was in good agreement with the experimental data and showed a gradual decrease of H3K9me3 in PCH with increasing demethylation activity (Fig 8C). Third, we evaluated the dependence of H3K9me3 in PCH on the concentration of origin sites (Fig 8D). We found an approximately linear response of the H3K9me3 levels up to the wild-type "*o*" site concentration range of approximately 30 μM.

From simulations of the kinetics of methyl mark propagation in virtual H3K9me3 induction experiments, we conclude that all transient perturbations on the minute time scale will not affect the overall PCH H3K9me3 levels due to the slow response of the system (Fig 8E). The simulations were started from the completely unmodified PCH state (e.g. all nucleosomes in state "*n*" and "*o*" sites at every 8th nucleosome), and the H3K9me3 level was followed over time. The steady-state PCH level of 38% was reached after approximately 33 h, corresponding to a propagation rate of 0.35 nucleosomes per hour from a given nucleation site. Notably, these kinetics are comparable to the value of 0.18 nucleosomes per hour measured experimentally for H3K9me3 propagation at the *POU5F1* promoter in mouse fibroblasts when inducing H3K9 trimethylation by tethering HP1α (Hathaway *et al*, 2012). One key parameter to modulate H3K9me3 levels in PCH is the SUV39H concentration (Fig 8F), which might vary between cell types. A 2-fold increase or decrease of the wild-type SUV39H concentration as observed between ESCs and MEFs would raise or lower H3K9me3 levels in PCH from an average value of 38% to about 51% or 28%, respectively.

In summary, we found that the predictions made by our model were in very good agreement with experiments. We conclude that it introduces an appropriate representation of the available data reported here and elsewhere that are relevant for the epigenetic network centered on H3K9me3 in mouse fibroblasts. The model quantitatively describes the properties of euchromatin and PCH determined here under different conditions and explains how a cell can robustly maintain the PCH state without the requirement to invoke additional components such as boundary proteins that bind to chromatin and block linear PCH spreading.

# Discussion

Here, we present a comprehensive analysis of the epigenetic network that silences the transcription of major satellite repeats in mouse fibroblasts. Based on data from advanced fluorescence microscopy methods and ChIP-seq, we explain how chromatin and protein interactions of MECP2, MBD1, DNMT1, SUV39H1, SUV39H2, JMJD2B/C, HP1α, HP1β, HP1γ, SUV4-20H1, and SUV4-20H2 are linked to 5meC, H3K9me3, and H4K20me3 modifications in PCH. From our quantitative analysis, we derive a predictive mathematical model that provides insight on how the silenced PCH state in fibroblasts is stably maintained and how it could be transmitted through the cell cycle. As mentioned above, H3K9me3 levels and transcriptional silencing in PCH vary between cell types (Supplementary Fig S2). The underlying mechanisms are only partly understood and beyond the scope of the present study. Nevertheless, our modeling framework provides an approach to evaluate the effect of key parameters. For example, differential regulation could rely on changing the composition of the nucleation complex (Fig 8B) or SUV39H abundance (Fig 8F). In addition, it is noted that SUV39H activity itself is controlled by posttranslational modifications like acetylation (Vaquero *et al*, 2007), methylation (Wang *et al*, 2013), or phosphorylation (Park *et al*, 2014), which links it to additional cellular pathways.

## Stably bound PCH complexes of SUV39H require the simultaneous presence of DNA methylation, MECP2, H3K9me3, and HP1

To rationalize our results, we propose that a network of interactions between 5meC, MECP2, MBD1, SUV39H, H3K9me3, and HP1 is responsible for the stably PCH-bound SUV39H complexes that are immobile on the time scale of minutes (Fig 4E, Table 1). Additional factors might further stabilize SUV39H as discussed below. Based on our AUC, FCCS, and F2H experiments, we conclude that most HP1 is present as a homo- or heterodimer in the cell and interacts with SUV39H1 that is in a monomer–dimer equilibrium (Fig 2, Supplementary Fig S5). This interaction is needed for stable SUV39H binding and H3K9 trimethylation in PCH as demonstrated in the triple knockdown of all HP1 isoforms (Fig 3A and B). The link between immobilized HP1 and SUV39H as well as the H3K9 trimethylation mark was further corroborated by FRAP measurements of HP1 in *Suv39h* dn cells, where H3K9me3 reduction and the loss of the binding partner SUV39H led to a strong increase in HP1 mobility and the loss of immobile HP1 (Supplementary Fig S4C) (Müller et al, 2009).

By quantifying parameters needed for PCH network modeling, our study provides an important extension of previously published FRAP studies of HP1 (Cheutin et al, 2003; Festenstein et al, 2003; Schmiedeberg et al, 2004; Dialynas et al, 2007). The concentration of immobilized HP1$\alpha$/$\beta$/$\gamma$ (1–3 $\mu$M, depending on cell cycle phase) and immobile SUV39H (~1.4 $\mu$M) in PCH measured here is compatible with an HP1 dimer interacting with a SUV39H dimer. The presence of PCH-bound SUV39H-HP1 complexes is consistent with our ChIP-seq analysis of SUV39H1, SUV39H2, HP1$\beta$, and H3K9me3, which were enriched at silenced but not at active intergenic/intronic major satellite repeats (Fig 4). These findings are in very good agreement with previous studies: (i) A SUV39H1 mutant lacking its N-terminal chromodomain (SUV39H1-$\Delta$N89) and thus being unable to interact with HP1 showed a homogeneous distribution throughout the nucleus and strongly reduced chromatin interactions in mammalian cell lines (Krouwels et al, 2005). (ii) HP1 dimerization was found to be important for maintaining H3K9me3 in yeast (Haldar et al, 2011). (iii) *In vitro* experiments showed that SUV39H1 interacts with HP1 by binding the molecular surface formed by dimerization of the chromoshadow-domain (Aagaard et al, 1999; Yamamoto & Sonoda, 2003; Nozawa et al, 2010).

Both HP1 and SUV39H1 are able to recognize H3K9me3 (Lachner et al, 2001; Jacobs & Khorasanizadeh, 2002; Jacobs et al, 2004), and the modification may confer some specificity for PCH binding via local H3K9me3 clusters, since each of the two chromodomains in an HP1 dimer may bind to one H3K9me2/3 residue (Thiru et al, 2004). Nevertheless, it is apparent from our quantification that H3K9me3 alone is not sufficient to stably and specifically tether the HP1-SUV39H complex to PCH. While high-affinity SUV39H binding was hardly present in euchromatin and the stably tethered SUV39H pool was enriched approximately 16- to 50-fold in PCH with respect to euchromatin, H3K9me3 levels were only moderately higher in PCH than in euchromatin (~1.4-fold). Rather, our quantitative chromatin interaction analysis suggests that SUV39H is tethered to PCH via interactions with multiple factors including MECP2, MBD1, and HP1 (Figs 1B, C and 2C, Supplementary Fig S5B). Since MECP2 binding is linked to 5meC (Nan et al, 1996), 5meC contributes to SUV39H immobilization. Further, H3K9me3 and 5meC are interrelated as

iMEF *Suv39h* dn cells showed reduced DNA methylation at major satellite repeats (Lehnertz et al, 2003; Fuks, 2005) although MECP2 and 5meC colocalized with PCH also in the absence of SUV39H (Supplementary Fig S2B). Notably, 5meC enrichment is not sufficient for PCH formation since a subset of intergenic major satellite repeats displays high 5meC levels but was devoid of SUV39H and transcriptionally active (Fig 4C and D). The above conclusions are supported by a number of previous findings: (i) Interactions of SUV39H1 with MECP2 have been demonstrated (Lunyak et al, 2002; Fujita et al, 2003). (ii) Krouwels et al showed that DNA demethylation increases SUV39H1 mobility and reduces the fraction of immobile SUV39H1, whereas HP1 mobility remained unchanged (Krouwels et al, 2005). This is compatible with our model since only 1–2% of total HP1 were present in the immobilized HP1-SUV39H complex. (iii) The loss of a DNMT1 complex reduced pericentric H3K9 methylation levels in human HeLa cells (Xin et al, 2004). (iv) The knockout of MECP2 in neuronal cells resulted in aberrantly low levels of H3K9me3 in PCH but not in euchromatin (Thatcher & LaSalle, 2006).

We conclude that the stability and specificity of the SUV39H-HP1-MECP2/MBD1 complex in PCH involves protein–protein interactions between these factors and some contribution from an increased 5meC level. Nevertheless, it is likely to be enhanced by additional protein factors. Since the PAX3 and PAX9 transcription factors were not stably bound at PCH (Supplementary Fig S1B and S4B), they are unlikely to play a direct role in stabilizing the HP1-SUV39H nucleation complex. Rather, PAX proteins might regulate the abundance of transcripts in PCH that potentially act as a binding platform for downstream factors (Maison et al, 2011). A number of other factors have been linked to PCH. For example, it was shown that the Mi-2/NuRD complex, which contains several interaction partners of SUV39H and HP1, is necessary to maintain the H3K9me3 mark in PCH (Sims & Wade, 2011). The contribution of such additional factors that enhance SUV39H binding in addition to MECP2 and/or MBD1 is implicitly considered in our quantitative model via the use of the experimentally determined enrichment of the SUV39H high-affinity binding sites in PCH that is independent of their exact molecular composition.

## Relatively sparse stably bound SUV39H nucleation complexes are sufficient to propagate H3K9me3 via chromatin looping

The stably bound SUV39H-HP1-MECP2/MBD1 complex in PCH was present at a concentration of approximately 1 $\mu$M as inferred from the concentrations of its constituting components of 1.4 $\mu$M (dimeric HP1 isoforms), 0.7 $\mu$M (dimeric SUV39H1 and SUV39H2), 1.4 $\mu$M (MECP2), and 0.2 $\mu$M (MBD1), while the nucleosome concentration in PCH was approximately 230 $\mu$M (Supplementary Table S7). Thus, bound SUV39H is sparsely distributed. We propose that the H3K9me3 modification is propagated via looping of the nucleosome chain to nucleosomes in spatial proximity from these complexes (Figs 6B and 8G). By ectopically tethering SUV39H1 to the nuclear lamina, we demonstrated experimentally that SUV39H immobilization indeed leads to the locally confined enrichment of H3K9me3 (Fig 6). Thus, chromatin-bound SUV39H might interact with substrate nucleosomes on the same chain or from another chromosome in spatial proximity. Our results are fully consistent with the experimental results from the DamID approach developed by van

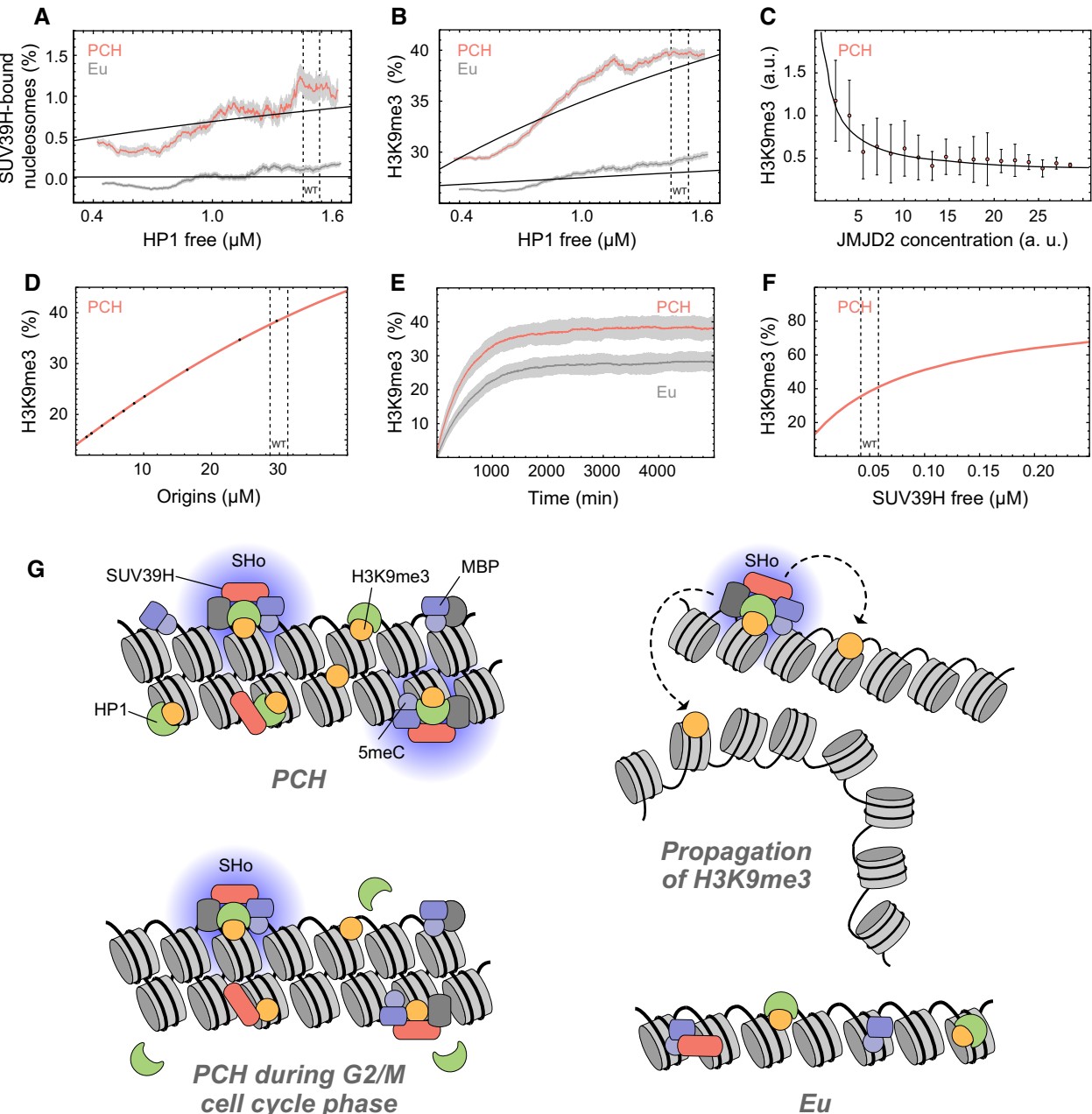

**Figure 8. Prediction of PCH features from network model.**

A   Dependence of SUV39H enrichment on HP1 concentration in PCH (red) and euchromatin (gray). Model predictions (black lines) were compared to the experimental values determined in the triple-knockdown experiments shown in Fig 3.

B   Same as panel A but for H3K9me3 instead of SUV39H.

C   Dependence of H3K9me3 levels on the concentration of the H3K9me3 demethylase JMJD2C. Error bars correspond to SD. The model prediction (solid line) agreed very well with the experimental data (points) derived from over-expression of JMJD2C-GFP.

D   Predicted average H3K9me3 levels in PCH versus concentration of the origins.

E   Stochastic time evolution of H3K9 trimethylation in PCH and euchromatin from a completely H3K9me3-deficient state. Red and dark gray traces are averages of 100 single stochastic trajectories with the standard deviation shown in light gray.

F   Dependence of H3K9me3 in PCH on SUV39H concentration. This parameter varies between different mouse cell types. For example, in ESCs the combined SUV39H1/2 concentration is 2-fold reduced compared to NIH-3T3 fibroblasts as estimated from their normalized RNA expression levels.

G   "Nucleation and looping" model for the propagation of H3K9me3 in PCH. The high-affinity binding sites with immobilized HP1 and SUV39H represent the SUV39H nucleation complex (Fig 4E; "SHo" in Fig 7B), which is highly specific for PCH. In contrast, the low-affinity binding sites composed of single protein factors were found throughout the whole nucleus, that is in both PCH and euchromatin. While soluble SUV39H proteins can methylate unmodified nucleosomes, the SUV39H nucleation complex provides a high local concentration of the enzyme and is responsible for the majority of catalytically productive collisions in PCH. Due to chromatin looping, the chromatin-bound SUV39H complexes can either methylate adjacent nucleosomes on the same chain or in 3D at other loci that reside in spatial proximity. The persistence of stably chromatin-bound SUV39H throughout the cell cycle (Fig 5) sustains the H3K9me3 modification.

Steensel *et al* that uses adenine methylation by DNA adenine methyltransferase (Dam) as readout for interaction. Both the extension of adenine methylation from chromatin-tethered Dam (van Steensel & Henikoff, 2000) as well as the extension of this DNA methylation from Dam tethered to the nuclear lamina (Kind *et al*, 2013) are in excellent agreement with our mechanism for setting H3K9me3 from SUV39H-bound sites on chromatin or the nuclear lamina.

We would like to emphasize that the propagation of H3K9me3 via the assembly of additional SUV39H-HP1-MECP2/MBD1 nucleation complexes is inherently limited in 3D without the requirement for additional insulator proteins: (i) Only sites that also have preexisting DNA methylation and bound MECP2/MBD1 in addition to H3K9me3 could lead to nucleation complex assembly. A newly formed H3K9me3 site alone would not be sufficient. (ii) The formation of productive nucleation sites is limited by the available amount of SUV39H. (iii) The H3K9 trimethylation activity originating from the nucleation complexes occurs only within the looping distance of chromatin around these complexes. Typical spreading distances for H3K9me3 limited by the diffusive motion of the chromatin fiber are 5–10 kb, in agreement with other experimental studies discussed elsewhere (Erdel *et al*, 2013). (iv) There is no preferred direction for diffusive motion of chromatin-bound SUV39H molecules and collisions rather occur in three dimensions with all nucleosomes in spatial proximity that could also be located on a different chromosome. This provides a straightforward explanation for the spherical shape of chromocenters, which may contain pericentromeric repeats from more than one chromosome (Probst & Almouzni, 2008).

### SUV39H bookmarks PCH during all cell cycle stages

According to our model, the reestablishment of H3K9me3 in PCH at newly assembled and unmodified histones after replication is mediated by SUV39H that remains bound throughout the cell cycle (Figs 5 and 8G, Supplementary Fig S1C). This process is likely to involve the SUV39H interaction partners MECP2 and MBD1 that show the same persistent binding enhanced by 5meC. This link between H3K9me3 and 5meC is in line with the previous report that the epigenetic inheritance of H3K9me3 involves DNA methylation (Hathaway *et al*, 2012). Furthermore, a relatively small fraction of HP1 was still immobilized at PCH during G2 phase in the FRAP experiments, which amounts to a significant concentration of approximately 1 μM due to the high level of total HP1. This is consistent with the detection of HP1α/β/γ in a quantitative proteomics analysis of mitotic chromosomes (Ohta *et al*, 2010), and the presence of HP1α and HP1γ at (peri-)centromeric chromatin on metaphase spreads (Nozawa *et al*, 2010; Hahn *et al*, 2013). In addition, SUV39H and HP1 were detected at nascent chromatin following DNA replication (Alabert *et al*, 2014). Thus, we conclude that the SUV39H nucleation sites persist during all phases of the cell cycle. The constitutive presence of the SUV39H enzymes at PCH together with the finding that SUV39H1 is sufficient to establish *de novo* H3K9me3 domains when immobilized at the nuclear lamina (Fig 6) has important mechanistic implications: It strongly suggests that the PCH-associated SUV39H molecules are responsible for locally reestablishing the H3K9me3 modification after its dilution during replication (Fig 8G). Thus, immobilized SUV39H molecules might act as bookmarking factors that stably transmit the PCH state through the cell cycle.

### H3K9me3 can robustly be maintained in PCH via the nucleation and looping mechanism

Linking modifications of histone residues to their readout by specific protein domains is an important aspect of current theoretical models that describe how epigenetic networks establish and maintain specific chromatin states based on dynamic nucleosome modifications (Dodd *et al*, 2007; Angel *et al*, 2011; Hathaway *et al*, 2012; Hodges & Crabtree, 2012). These include positive feedback loops where modified histones (directly or indirectly) recruit enzymes that catalyze a similar modification on nearby nucleosomes. One class of these models is characterized by relatively robust bistable chromatin states that can stably co-exist for a certain set of conditions (Dodd *et al*, 2007; Angel *et al*, 2011). This originates from the presence of multiple positive feedback loops as well as "long-range" interactions along the nucleosome chain. To limit the spreading of a distinct modification to chromatin outside the domain under consideration, the existence of boundary factors is invoked. If H3K9me3 was able to spread within PCH via such a mechanism, one would have to explain how spreading is confined in 3D for the 28 ± 1 chromocenters per nucleus with an average volume of 2.69 ± 0.04 μm$^3$ or approximately 6 Mbp of DNA (Cantaloube *et al*, 2012). Accordingly, the cell would have to maintain a rather elaborate spherical boundary structure for the 3D confinement of H3K9me3 to PCH, for which there is no evidence. It is also noted that for H3K27me3, the view that boundary factors like CTCF are required to limit domain spreading along the nucleosome chain has been challenged by two recent studies (Schwartz *et al*, 2012; Van Bortle *et al*, 2012). Nevertheless, the "nucleation and looping" mechanism proposed here would be fully compatible with the function of insulators as architectural factors that confine the 3D organization of chromatin by establishing interactions between distant sites that would promote or inhibit long-range contacts between nucleosomes and chromatin-bound epigenetic modifiers (Ong & Corces, 2014). Furthermore, we measured that H3K9me3 was less than 2-fold reduced in euchromatin, which suggests that H3K9me3-dependent feedback loops are rather weak. For strong feedback, it would be difficult to explain why the mark would not spread throughout the rest of the genome via the same mechanism as in PCH. Finally, we did not find evidence for bistable H3K9me3 states when perturbing the balance between H3K9me3 methylation and demethylation (Figs 3A, B and 8A–C). Rather, the H3K9me3 distribution obtained from measurements of single cells in dependence of the HP1 concentration showed a gradual transition from wild-type levels to those measured in *Suv39h* dn cells (Fig 3B).

Recently, an alternative model derived from experiments in which HP1α was recruited to the *POU5F1* promoter to induce heterochromatin formation and gene repression was introduced (Hathaway *et al*, 2012; Hodges & Crabtree, 2012). The experimentally determined H3K9me3 domain with smoothly decreasing borders was modeled with a 1D-lattice model for a chain of 257 nucleosomes, in which the modification is propagated by nearest-neighbor interactions from the nucleation site along the chain. Here, we detected the endogenous equivalent of these ectopic nucleation sites for high-affinity binding of SUV39H complexes in PCH. However, the constraints on protein–chromatin interactions and protein concentrations imposed from our experiments were not compatible with forming stable H3K9me3 domains via linear

nearest-neighbor spreading. Furthermore, it provided insufficient specificity with respect to the presence/absence of the high-affinity nucleation sites found in PCH versus euchromatin. In contrast, our nucleation and looping model schematically depicted in Fig 8G was found to be robust with respect to maintaining the experimentally measured PCH features: (i) It provides an intrinsic limit for the SUV39H-dependent extension of H3K9me3 within the system. (ii) Stochastic number fluctuations of cellular factors have little effect (Fig 7C and D). (iii) Perturbations of SUV39H1 binding (Fig 8A), histone demethylation activity (Fig 8C), or the concentrations of SUV39H high-affinity sites (Fig 8D) change the H3K9me3 level only gradually and in a reversible manner. (iv) Transient perturbations of the system in the range of minutes were insignificant, since the H3K9me3 propagation rate is slow with only 0.35 nucleosomes per hour (Fig 8E).

Furthermore, the response of the quantitative model (Fig 7B) toward perturbations was tested experimentally by HP1 knockdown and overexpression of the histone demethylase JMJD2C. In these experiments, a gradual change of the H3K9me3 level depending on HP1 (Figs 3A and 8A, B) or JMJD2C (Fig 8C) concentration was measured, which was reproduced by the model. Likewise, the steady-state methylation level was not bistable since the looping-mediated propagation rate of H3K9me3 decreases approximately linearly with the SUV39H occupancy at the nucleation site. This is consistent with the gradual reduction of GFP expression from the activated *POU5F1* promoter observed in MEFs after triggering HP1 recruitment (Hathaway *et al*, 2012).

## Concluding Remarks

Our findings lead us to propose that in mouse fibroblast cells the PCH state is maintained by a nucleation and looping mechanism, in which the H3K9me3 modifications originate from relatively sparsely distributed nucleation sites of stably bound SUV39H complexes (Fig 4E). The local extension of the H3K9me3 modification occurs via looping of the nucleosome chain to mediate methylation of nucleosomes by a chromatin-bound HP1-SUV39H complex in spatial proximity against an unspecific demethylation activity provided by JMJD2 enzymes (Fig 8G). This mechanism is site-specific and robust toward number fluctuations of its components. SUV39H–chromatin complexes persisted through the cell cycle and could act as bookmarking factors for memorizing PCH silencing of transcription (Fig 5). Our model lacks bistable states and it does neither involve nearest-neighbor feedback loops for linear spreading of H3K9me3 nor the presence of locus-specific boundary factors to limit such a process. The predicted behavior of the system according to the nucleation and looping model in response to perturbations was in excellent agreement with the experimental findings. Additionally, it is well suited to rationalize general features of cellular systems that establish, maintain, or modulate epigenetic patterns of characteristic domain size (Erdel *et al*, 2013). The proposed mechanism is fully consistent with results of studies on the distribution of H3K9 methylation along the nucleosome chain upon chromatin-tethering of HP1 (Hathaway *et al*, 2012) or the yeast homologue of SUV39H (Kagansky *et al*, 2009), as well as with the shape of the H3K27me3 domain observed in *Arabidopsis* that is involved in silencing the floral repressor FLC (Angel *et al*, 2011). Thus, its

conceptual features might be relevant for heritable functional chromatin states at other genomic loci.

## Materials and Methods

### Cell lines

Experiments were conducted with GFP and RFP constructs in the murine NIH-3T3 fibroblast cell line or in immortalized mouse embryonic fibroblasts (iMEF) wild-type and mutant cell lines (Peters *et al*, 2001; Schotta *et al*, 2008; Müller *et al*, 2009). Autofluorescent proteins were either expressed as stable (inducible) cell lines or introduced via transient transfection as described in the Supplementary Materials and Methods.

### Fluorescence microscopy imaging, FRAP, and FCS

Confocal imaging, FRAP and FCS experiments, and associated data analysis were conducted with a Leica TCS SP5 or Zeiss LSM 710 confocal laser scanning microscope as described previously (Müller *et al*, 2009; Erdel *et al*, 2010) and in the Supplementary Materials and Methods. Immunofluorescence was conducted with primary anti-H3K9me3 (Millipore, Abcam ab8898), anti-HP1α (Euromedex, 2HP-1H5-AS), anti-HP1β (Euromedex, 1MOD-1A9-AS), anti-HP1γ (Euromedex, 2MOD-1G6-AS) or anti-H4K20me3 (Abcam, ab9053) antibodies and a secondary goat anti-rabbit/mouse Alexa 568 antibody or anti-rabbit/mouse Alexa 633 antibody (Invitrogen, Molecular Probes).

Protein enrichments and H3K9 trimethylation levels were measured from high-resolution microscopy images using the ImageJ software as described in the Supplementary Materials and Methods. FRAP measurements were fitted either to a diffusion model, a binding model or a reaction-diffusion model that incorporates both diffusion and binding processes. The data from the model that yielded the best fit was used for further analysis and modeling (Müller *et al*, 2009).

### Protein interaction analysis by FCCS, F2H, ChIP-seq and AUC

Protein–protein interaction analysis of the soluble nuclear fraction in living cells was done by FCCS (Erdel *et al*, 2010), and interactions of chromatin-bound proteins were measured via a fluorescent two-hybrid assay (F2H) in the cell nucleus as reported before (Chung *et al*, 2011). ChIP-seq experiments were conducted as described previously (Teif *et al*, 2012); the data produced have been deposited to the GEO database (accession number GSE58555). Measurements of HP1 association states with recombinant proteins were performed by analytical ultracentrifugation according to the workflow given in our previous work (Kepert *et al*, 2003; Fejes Tóth *et al*, 2005). Details on all methods and associated data analysis are given in the Supplementary Materials and Methods section.

### Network modeling

The model of the epigenetic network was calculated for a chromatin fiber of 300 nucleosomes. Each nucleosome on the fiber was able to collide with others via chromatin looping following the collision

probability determined by the local concentration of one nucleosome in the proximity of the others as described previously (Rippe *et al*, 1995; Rippe, 2001; Erdel *et al*, 2013). The resulting values for an increased local concentration of nucleosomes in the proximity of the first nucleosome at the 0-position due to chromatin looping is shown in Fig 7A with the concentration of high-affinity binding sites ("origins") measured as described in the text. The model consists of a system of ordinary differential equations (ODEs). The core variables constituting the network (Fig 7B) are the local probability of methylation and the local probabilities of occupation by HP1, SUV39H, and HP1-SUV39H complex that depend on time and the position of nucleosomes and nucleation origins on DNA. The model variables, parameters, and reference values are summarized in Supplementary Table S7. For the euchromatin fiber system, the ODEs lack the nucleation origins. Stochastic kinetic traces and stochastic steady-state distributions were simulated with the Gillespie stochastic simulation algorithm (Gillespie, 2007) implemented in C++. The state of each nucleosome was derived numerically based on the deterministic formalism implemented in Mathematica 9.0 (Wolfram Research). See Supplementary Materials and Methods for details on model implementation and fitting.

**Supplementary information** for this article is available online: http://msb.embopress.org

## Acknowledgements

We are grateful to Malte Wachsmuth, Vladimir Teif, Barna Fodor, and Nick Kepper for help and discussions and thank Thomas Jenuwein, Adrian Bird, Natasha Murzina, and Ken Yamamoto for plasmid vectors and cell lines. The fluorescence microscopy work was conducted in the DKFZ Microscopy Core Facility, the Nikon Imaging Center at the University of Heidelberg, and the EMBL Advanced Light Microscopy Facility. The project was supported by the BMBF projects EpiSys (0315502) and CancerEpiSys (0316049). Additional funding of work in the lab of GS was via a DFG grant within SFB1064.

## Author contributions

KMO, FE, TH, and KR designed the study. KMO, JPM, AR, MH, CB, QZ, and SK acquired the experimental data. KMO, JPM, AR, MH, AM, QZ, SK, GS, TH, and KR analyzed and interpreted the data. AM, TH, FE, and KR conducted calculations and mathematical modeling. Manuscript writing was done by KMO, FE, AR, AM, TH, and KR with contributions from all authors regarding critical revisions.

## Conflict of interest

The authors declare that they have no conflict of interest.

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
