## [Review Process File · Molecular Systems Biology]

Specificity, propagation and memory of pericentric heterochromatin

Katharina Müller-Ott, Fabian Erdel, Anna Matveeva, Jan-Philipp Mallm, Anne Rademacher, Matthias Hahn, Caroline Bauer, Qin Zhang, Sabine Kaltofen, Gunnar Schotta, Thomas Höfer, Karsten Rippe

Corresponding author: Karsten Rippe, DKFZ & BioQuant Center

Review timeline:	Submission date:	19 April 2014
	Editorial Decision:	14 May 2014
	Revision received:	13 June 2014
	Editorial Decision:	20 June 2014
	Revision received:	10 July 2014
	Accepted:	15 July 2014

Editor: Maria Polychronidou

Transaction Report:

1st Editorial Decision

14 May 2014

Thank you again for submitting your work to Molecular Systems Biology. We have now heard back from the three referees who agreed to evaluate your manuscript. As you will see from the reports below, the referees acknowledge that the presented analysis is potentially interesting. However, they raise a series of concerns, which should be carefully addressed in a revision of the manuscript.

Without repeating all the points listed below, most of the comments refer to the need to better document and discuss/clarify several points throughout the manuscript. In particular, the reviewers point out that the manuscript needs to be streamlined and the novel findings should be better placed in the context of the current knowledge and previously published work. Moreover, reviewer #1 expresses concerns related to the mathematical modeling while reviewers #2 and #3 mention some issues concerning data presentation (i.e. regarding the images shown in Fig. 1).

REFeree REPORTS:

Reviewer #1:

This is an interesting manuscript analysing pericentric heterochromatin in fibroblasts from both an experimental and a computational angle. Much of the work is thorough and offers an unusually complete analysis of the systems' dynamics that will undoubtedly be useful elsewhere. However, much of this good work is undone by some rather overblown claims, mostly in the Discussion, which need to be toned down. Furthermore, the authors need to be much more precise in their

cavalier use of epigenetics terminology. The use of computational modelling also needs to be made more decisive. Nevertheless, with appropriate modifications, the manuscript could make a useful contribution to the literature. Specific points are listed below:

* One of my main objections to the manuscript as it stands is in the inappropriate use of epigenetic terminology. Epigenetics refers to mitotic heritability that is not dependent on DNA sequence. Yet at heart the phenomenon studied here is not really epigenetic, as it relies fundamentally on CpG methylation that are capable of renucleating the Suv39h and H3K9me3 via 5meC. Indeed the authors own simulations (Fig 8F) make exactly this point. The authors need to correct this error and point out much more clearly the fundamental basis for the PCH state, namely CpG-mediated 5meC, recruiting MeCP2, which then induces origin sites and nucleation of Suv39h, which mediates H3K9me3 spreading via chromatin looping and an HP1/H3K9me3 feedback loop. Claims that it is the Suv39h that is fundamentally the epigenetic memory element defining the PCH state (p4 bottom, p26 top) are misleading. Nor do the authors address the origin of epigenetic memory (p4).

* The authors claim, as one of the main features of their model, that it explains the presence of PCH without the need to invoke boundary elements. As far as I can see, the PCH is defined by a higher density of origin sites, with the PCH boundary defined by a change in origin site density. In this sense, the PCH boundary is hardwired into the model fundamentally through an altered CpG density. This needs to be made clear.

* The discussion on p11 of Suv39h being preferentially active in PCH is hard to follow. Moreover the title seems to imply that the Suv39h is somehow more active whereas it seems that it is the concentration that is being modulated.

* The authors claim that their system is not bistable and present convincing evidence to back this up. This is then compared to prior modelling which emphasised the bistable nature of epigenetic states and suggest that their model may be widely applicable. In my view this claim is problematic, since, as pointed out above, the pericentric heterochromatin system is not genuinely epigenetic. Other systems dictating developmental state must be epigenetic due to the absence of DNA sequence variation. Furthermore, as I understand it, the FLC system is also very different, as there it is environmental exposure that is memorised in cells that are otherwise identical. Only in these cases, where the memory really is epigenetic, is there a requirement for bistability.

* Currently I am unconvinced by the complexity of the mathematical modelling. It seems that the authors get out of the modelling more or less what they put in. Also the lack of an opposing histone modification more or less guarantees a lack of bistability. However, I think there is scope to make the modelling more decisive, and potentially more simple. For example, it is not clear to me what determines the decay length of the H3K9me3 levels away from origin sites. Is this length scale dictating by the chromatin looping (Fig 7A) or through the H3K9me3/HP1 feedback loop?

Reviewer #2:

In this manuscript, Rippe and colleagues developed a quantitative model to explain pericentric heterochromatin (PCH) formation and propagation. They performed immunofluorescence measurements, fluorescence bleaching experiments (FRAP), fluorescent correlation spectroscopy (FCS), fluorescent two-hybrid measurements, ChIP-seq experiments, and perturbations of the key factors involved (e.g. HP1 proteins, Suv39h1/3, etc.). Their model explains most published observations, is robust to perturbations, and does not necessitate insulators to halt heterochromatin propagation.

This is an agonizingly long paper with so many details that the big picture is often lost. The study overall is well-developed and proposes a superior model to explain PCH formation and propagation but which is obscured by too many details. Also of note, many of the authors' observations are not entirely novel, including good parts of Figures 1-3, although it could be argued that they were reproduced here for a greater cause. Many of the paragraphs end with concluding statements that are not novel and/or expected. The major caveat though is the fact that while the model nicely explains many of the PCH features, including, importantly, robustness and spatial confinement, it does not take into consideration and hence does not explain the extensive use of insulators in the mammalian

genome.

Additional points:

In Figure 1 the authors only provide DAPI for some of the factors and not others. This is confusing, especially since some do not appear to be quite overlapping, for example MeCP2 and DAPI seem to appear as bigger PCH foci while Mbd1 and 5meC seem to be numerous and smaller. Mbd and 5meC should be shown together to support the authors' claim. In the same Figure, the diffuse proteins do not appear in nucleoli, as expected, except for Jmjd2b which seems to be all over the nucleus. Is this an artefact of some sort? Why is it not excluded from nucleoli? Also, in the lower panel the DAPI shows 2 cells while the MeCP2 shows only one of them. How is this possible?

From the ChIP-seq experiments, the authors conclude "...the simultaneous presence of HP1, H3K9me3 and 5meC" (p.12). While logical and likely correct, it cannot be postulated from population based experiments. ChIP-re-chip might solve this issue.

The cell cycle experiments are incomplete and have only been conducted on a fraction of the factors. For example, the authors state: "Localization of the DNA methyltransferase Dnmt1 to PCH was cell cycle-dependent" - Where is it shown?

The authors use PCH and satellite repeats interchangeably, but satellites are just a part of the story.

The authors predict sparse binding of Suv39h every ~170 nucleosomes (p.13-14). I wonder whether this prediction is testable.

Minor:

Expanded view pages 33-37 are in a wrong order

P.5 line 19: H4K20 and Suv4-20h are not shown in Figure 1A but in Figure E1

Reviewer #3:

In this manuscript the authors characterized several quantitative parameters of factors enriched at pericentric heterochromatin domains in mouse cells in order to define a model, which describes how these factors contribute to the stability and confinement/compaction of these domains. These factors include proteins, DNA methylation and histone marks with emphasis on the H3K9me3 modification, which is the hallmark of these heterochromatin domains.

They used a combination of advanced microscopy techniques including imaging, FRAP and FCS to determine dynamics and concentration of these factors at pericentric domains. They also used available data from the literature and analytical ultracentrifugation to investigate HP1 dimerization dissociation constant.

They then integrated all these parameters to generate a model according to which, sparse nucleation sites containing the H3K9 HKMT Suv39H propagates by looping the H3K9me3 modification to nucleosomes in close spatial proximity to maintain pericentric heterochromatin status.

Taken together this manuscript represents a significant amount of work, which proposes a model that should be of interest for the scientific community in the fields of nuclear organization and genome stability. However there are some points that should be considered (1) to improve the clarity of the manuscript, and (2) to cite some original work that reported same findings earlier than those described in this manuscript. Indeed several times in the manuscript, it is not clear for a non-specialist reader to discriminate novel findings from confirmation of already reported data. The added value of the work reported here is the modeling aspect and the characterization of numerical value for protein concentration/enrichment or association constant. Authors should thus concentrate on this aspect.

Below are some specific comments that authors should consider

1) Some of the data presented in the manuscript are known for long time. For example the description of accumulation of HP1H3K9me3, Suv39h, MeCP2 and HP1 at pericentric domains in figure 1A along with the corresponding text could be shortened.

2) The fact that in Suv39h dn cells transcription of satellite sequences increase was published (Lenhertz et al., 2003 Current Biol.; Martens et al., 2003 Embo Journal)

- 3) Decondensation of chromocenters upon deacetylase inhibitor was reported in (Taddei et al., 2001 Nat Cell Biol.)
- 4) Dynamics of HP1 determined by FRAP analysis was investigated in Festenstein et al., 2003, Science; Cheutin et al., 2003 Science; Schmiederberg et al., 2004 Molecular Biology of the Cell ; Dialynas et al., 2007 Journal of Cell Science. In this respect authors should discuss their own results in light of those published.
- 5) Presence of Suv39H1 on mitotic chromosome was also reported in Melcher et al., Mol. Cell. Biol.
- 6) It is not clear whether every data reported here have been generated with cell lines that stably express the fluorophores or cell lines that are transiently transfected. This should be indicated in the figure legend for every figure showing quantification of enrichment and immobilized protein.

1st Revision - authors' response

13 June 2014

Reviewer #1

“This is an interesting manuscript analysing pericentric heterochromatin in fibroblasts from both an experimental and a computational angle. Much of the work is thorough and offers an unusually complete analysis of the systems' dynamics that will undoubtedly be useful elsewhere. However, much of this good work is undone by some rather overblown claims, mostly in the Discussion, which need to be toned down. Furthermore, the authors need to be much more precise in their cavalier use of epigenetics terminology. The use of computational modelling also needs to be made more decisive. Nevertheless, with appropriate modifications, the manuscript could make a useful contribution to the literature. Specific points are listed below:”

We thank this reviewer for appreciating the high quality of our work. We clarified the claims in the Discussion and carefully revised the manuscript with attention on epigenetic terminology. Specific points are addressed below.

“One of my main objections to the manuscript as it stands is in the inappropriate use of epigenetic terminology. Epigenetics refers to mitotic heritability that is not dependent on DNA sequence.”

A number of definitions have been put forward and the reviewer uses a definition of epigenetics that is different from ours. We have clarified this issue in the introduction of the revised manuscript by referring to the following definition for ‘epigenetics’, which we consider to be well-accepted in the field: “An epigenetic trait is a stably heritable phenotype resulting from changes in a chromosome without alterations in the DNA sequence.” (Berger et al, 2009). We note that many current papers call essentially any histone modification 'epigenetic', even though a mechanism for transmitting histone modifications through replication and their association with a stably heritable phenotype has only been shown for a few cases. That being said, H3K9me3 along with H3K27me3 and H4K20me1 are considered well established 'epigenetic' histone modifications (Bonasio et al, 2010). Following the above definition we, like the majority of researchers in the field of epigenetics, consider DNA methylation a 'true' epigenetic modification despite its inherent sequence dependency (only CpGs can be methylated). However, according to the Berger et al. definition of epigenetics, an epigenetic state is not required to be independent of DNA sequence but only to occur without alterations of the DNA sequence. The 5meC modification can be set and removed without altering the DNA sequence, a given methylation pattern is copied through the cell cycle via DNMT1 and this pattern can be associated with a heritable phenotype.

“Yet at heart the phenomenon studied here is not really epigenetic, as it relies fundamentally on CpG methylation that are capable of renucleating the Suv39h and H3K9me3 via 5meC. Indeed the authors own simulations (Fig 8F) make exactly this point. The authors need to correct this error and point out much more clearly the fundamental basis for the PCH state, namely CpG-mediated 5meC, recruiting MeCP2, which then induces origin sites and nucleation of Suv39h, which mediates H3K9me3 spreading via chromatin looping and an HP1/H3K9me3 feedback loop.”

Following the above definition of epigenetics, we do not consider it an error to call the PCH system ‘epigenetic’ since it exhibits the following feature: The expression of major satellite can be largely different for different mouse cell types with identical genomes such as embryonic stem cells, *in vitro* differentiated cells derived from these, mouse embryonic fibroblasts and embryonic trophoblasts

(see Figure E2A in our revised manuscript which now compares transcription levels of embryonic stem cells and mouse embryonic fibroblasts, as well as (Lehnertz et al, 2003; Martens et al, 2005)). The PCH domain can switch from an ‘active’ state with low H3K9me3 and high levels of major satellite repeat transcription to an ‘inactive’ state that exhibits elevated H3K9 trimethylation and has transcription of major satellite repeats shut down (see also Figure 6 in (Martens et al, 2005)). Both states are heritable and are stably transmitted through mitosis in the respective cell types. Since the DNA sequence of the domain does not change, while the functionality does, this is prototypic epigenetic behavior, which has also been recognized previously by others (Probst & Almouzni, 2008).

Concerning the role of DNA methylation in establishing PCH, we have clarified its contribution in mouse fibroblasts in the revised manuscript. The CpG density in the major satellite consensus sequence is increased approximately 3-4-fold over genome average while DNA methylation levels are similar to the genome average (approximately 85%) yielding a 3-4-fold increase of 5meC. However, this increase of 5meC alone is not sufficient to induce the PCH state and SUV39H-dependent H3K9me3 transcriptional silencing. Furthermore, it is noted that the CpG distribution in the genome is not homogenous and CpG cluster into CpG islands (CGIs). About 16,000 CGIs are annotated in the UCSC Genome Browser for the mouse mm9 assembly that have a typical CpG content of about 10% (i. e. 10-fold enriched over average) but most of them are unmethylated (see (Teif et al, 2014) and references therein). Thus, these CGIs contain even higher CpG density than in PCH, but remain unmethylated and are not silenced like PCH, i.e. higher CpG density is not sufficient to induce transcriptional silencing via histone methylation/the PCH machinery. Furthermore, the intergenic major satellite repeats that can be uniquely mapped (Figure 4) have a CpG density of 2 ± 0.8 % and a DNA methylation level of 87 ± 7 %. As inferred from H3K36me3 ChIP-Seq, 12 repeats were silenced and carried the H3K9me3 mark while two repeats were transcribed and lacked H3K9me3 without a correlation with their 5meC levels. We conclude that neither having a high CpG density as in CGIs nor having a 3-4 fold increase of 5meC over genome average is sufficient to induce transcriptional silencing via H3K9me3. Accordingly, our PCH model involves 5meC and subsequent binding of MECP2 and MBD1 to stabilize the SUV39H nucleation site by protein interactions with SUV39H and/or HP1. Consistent with this view, the immobile MECP2 fraction was 80-fold higher in PCH, which cannot be rationalized by the higher 5meC level but indicates that it is part of the SUV39H1-HP1 complex where its chromatin binding is stabilized by protein-protein interactions.

“Claims that it is the Suv39h that is fundamentally the epigenetic memory element defining the PCH state (p4 bottom, p26 top) are misleading. Nor do the authors address the origin of epigenetic memory (p4).”

This point is well taken and we agree that it is debatable whether persistent SUV39H binding is the origin of the ‘epigenetic memory’ or a consequence of other PCH features. Accordingly, we have changed the term ‘epigenetic memory’ into memory or chromatin ‘bookmarking’ (a term used in previous studies for proteins that remain stably bound to chromatin) to describe the stable binding of SUV39H throughout the cell cycle. Furthermore, we have added FRAP experiments of SUV39H1 in mitotic cells to quantitate its chromatin interactions in Fig. 4. We find that most SUV39H1 remained stably bound, which corroborates our previous conclusions.

“The authors claim, as one of the main features of their model, that it explains the presence of PCH without the need to invoke boundary elements. As far as I can see, the PCH is defined by a higher density of origin sites, with the PCH boundary defined by a change in origin site density. In this sense, the PCH boundary is hardwired into the model fundamentally through an altered CpG density. This needs to be made clear.”

With ‘boundary’ we refer to *trans*-acting factors like CTCF that are frequently postulated to limit the spatial extension of chromatin domains. This is now explained in the revised manuscript.

“The discussion on p11 of Suv39h being preferentially active in PCH is hard to follow. Moreover the title seems to imply that the Suv39h is somehow more active whereas it seems that it is the concentration that is being modulated.”

We rephrased this section of the manuscript. According to our model, the SUV39H enzymatic activity is indeed invariant and does not depend on the compartment it resides in. The activity increase of the SUV39H pool in PCH is solely due to the higher effective concentration of bound SUV39H1.

“The authors claim that their system is not bistable and present convincing evidence to back this up. This is then compared to prior modelling which emphasised the bistable nature of epigenetic states and suggest that their model may be widely applicable. In my view this claim is problematic, since, as pointed out above, the pericentric heterochromatin system is not genuinely epigenetic. Other systems dictating developmental state must be epigenetic due to the absence of DNA sequence variation. Furthermore, as I understand it, the FLC system is also very different, as there it is environmental exposure that is memorised in cells that are otherwise identical. Only in these cases, where the memory really is epigenetic, is there a requirement for bistability.”

As explained above we consider PCH a prototypic epigenetic system that can change expression levels of RNA transcribed from major satellite repeats in the absence of DNA sequence alterations via H3K9me3-mediated silencing. For a given cell type, the corresponding expression phenotype is stably transmitted through cell division. Thus, we do not follow the line of argumentation put forward by the reviewer that there are systems where the “memory really is epigenetic” with a requirement for bistability, while the PCH state would not be epigenetic.

Furthermore, we feel it justified to discuss that chromatin-bound enzymes immobilized at nucleation sites may establish an activity gradient in conjunction with a spatial confinement for this activity also in other chromatin domains. Notably, experimentally determined patterns around artificially tethered enzymes are fully consistent with our ‘nucleation and looping’ model as discussed in detail in our previous theoretical work (Erdel et al, 2013). In addition, such a mechanism fits well with the extend of H3K27me3 domains measured experimentally for H3K27me3-mediated silencing in Arabidopsis (Angel et al, 2011) and in Drosophila (Comet et al, 2011). For other modifications, the specific nature of the nucleation sites and the molecular details might be different. Nevertheless, we feel that the existing experimental data are in support of the ‘nucleation and looping’ mechanism that we propose in our study. Thus, we should be given some freedom to point out that this mechanism might be operational outside of the specific system we have studied. In the revised manuscript, we clarified that this conclusion is based on evaluating existing data for other systems, which have been discussed previously (Erdel et al, 2013).

“Currently I am unconvinced by the complexity of the mathematical modelling. It seems that the authors get out of the modelling more or less what they put in. Also the lack of an opposing histone modification more or less guarantees a lack of bistability. However, I think there is scope to make the modelling more decisive, and potentially more simple.

We would like to point out that an opposing histone modification is not required to obtain bistability. Cooperative binding of SUV39H to methylate H3K9 would be sufficient. We believe that the present model provides the simplest account for the experimental data in the paper. To make this point clearer, we have amended and somewhat simplified the relevant text on pages 14 and 15. In particular, the model focuses on the signature histone modification of PCH, i. e. H3K9me3, and the interacting proteins HP1 and SUV39H. The parameterization of the model accounts for all occupancy states at their measured frequencies. Specifically, since the demethylation rate of H3K9me3 is taken from experimental measurement, the methylation rate is constrained by the experimentally determined H3K9me3 enrichment in heterochromatin and the frequency of HP1 and SUV39H bound states that shield H3K9me3 from demethylation. Based on these experimental constraints, we find that the model accounts for heterochromatic H3K9me3 density solely through the local enrichment of SUV39H by origin sites. No further mechanisms, such as autocatalysis and local spreading of H3K9me3 along the chromatin fiber, are required. Thus the model shows that our proposed mechanistic interpretation of PCH maintenance is quantitatively consistent with the data.

“For example, it is not clear to me what determines the decay length of the H3K9me3 levels away from origin sites. Is this length scale dictating by the chromatin looping (Fig 7A) or through the H3K9me3/HP1 feedback loop?”

The decay length is indeed determined by chromatin dynamics/looping. Based on the parameters we measured experimentally, we found that the H3K9me3/SUV39H feedback loop is too weak to induce significant spreading of H3K9me3 along the nucleosome chain. The scope of our model is to integrate the quantitative description of protein concentrations and interactions and to predict the stability of the system upon different perturbations, including dilution during cell cycle progression. In principle, stronger H3K9me3/SUV39H feedback could lead to bistability, so the observed monostability is not hardwired into the model. Thus, the (qualitative and quantitative) system response predicted by the model is not put in *a priori*. We find it truly insightful to understand how PCH is stably maintained through replication and in case of protein concentration fluctuations (compare Fig. 7).

Reviewer #2

“In this manuscript, Rippe and colleagues developed a quantitative model to explain pericentric heterochromatin (PCH) formation and propagation. They performed immunofluorescence measurements, fluorescence bleaching experiments (FRAP), fluorescent correlation spectroscopy (FCS), fluorescent two-hybrid measurements, ChIP-seq experiments, and perturbations of the key factors involved (e.g. HP1 proteins, Suv39h1/3, etc.). Their model explains most published observations, is robust to perturbations, and does not necessitate insulators to halt heterochromatin propagation.

This is an agonizingly long paper with so many details that the big picture is often lost. The study overall is well-developed and proposes a superior model to explain PCH formation and propagation but which is obscured by too many details. Also of note, many of the authors' observations are not entirely novel, including good parts of Figures 1-3, although it could be argued that they were reproduced here for a greater cause. Many of the paragraphs end with concluding statements that are not novel and/or expected.”

We did our best to make the revised paper entertainingly short. However, due to the design and the scope of our study there is a limit to what we can accomplish in this respect, since we have acquired a large amount of data to derive a systems description of a prototypic epigenetic network. For the purpose of our study, it was necessary to quantify the abundance and interactions of the factors related to pericentric heterochromatin (PCH) since most previous studies rely only on qualitative descriptions as recognized by the reviewer. One example is the colocalization with chromocenters that was previously observed for many proteins: Without quantitation this might either reflect PCH-specific binding or simply unspecific chromatin binding if the enrichment follows the chromatin density that is increased ~2-fold in chromocenters. To understand the differences between euchromatin and PCH, it is crucial to distinguish those scenarios. To clarify our focus on a quantitative and comprehensive description, we have replaced Fig. 1A with a scheme of our approach that emphasizes the quantitative analysis. Furthermore, we have carefully revised all subsections of Results section to clarify the question addressed, the state of relevant previous knowledge and the novel insight provided by our work.

“The major caveat though is the fact that while the model nicely explains many of the PCH features, including, importantly, robustness and spatial confinement, it does not take into consideration and hence does not explain the extensive use of insulators in the mammalian genome.”

We agree that insulator or boundary elements are used in numerous studies to explain the confinement of chromatin states with specific epigenetic modifications or the targeting of enhancer-mediated gene activation to a certain promoter. However, the wide-spread use of this model does not guarantee its correctness. In fact, the few critical experimental tests of the relation of insulator elements and spreading of histone modification marks conclude (e. g. for H3K27me3 domains in *Drosophila*) that “these observations argue against the concept of a genome partitioned by specialized boundary elements and suggest that insulators are reserved for specific regulation of selected genes” (Schwartz et al, 2012). Another study found that the knockdown of the prototypic insulator protein CTCF induced loss of the H3K27me3 modification within a domain rather than its spreading beyond the wild-type domain boundaries (Van Bortle et al, 2012). Thus, we consider it a strength and not a weakness of our model that it does not need to invoke the existence of boundary/insulators to confine H3K9me3.

Additional points:

In Figure 1 the authors only provide DAPI for some of the factors and not others. This is confusing, especially since some do not appear to be quite overlapping, for example MeCP2 and DAPI seem to appear as bigger PCH foci while Mbd1 and 5meC seem to be numerous and smaller. Mbd and 5meC should be shown together to support the authors' claim. In the same Figure, the diffuse proteins do not appear in nucleoli, as expected, except for Jmjd2b which seems to be all over the nucleus. Is this an artefact of some sort? Why is it not excluded from nucleoli? Also, in the lower panel the DAPI shows 2 cells while the MeCP2 shows only one of them. How is this possible?

A careful survey of the sizes of PCH foci for different proteins yielded no significant differences. Thus, the cells shown in the previous Fig. 1A (now Fig. E1) are not representative in terms of differences in PCH foci size. In general, proteins that localize to PCH foci do so for all foci present in a given cell and also co-localize with each other in chromocenters. This is now stated in the figure legends in the revised version of the manuscript. The fact that JMJD2B is not excluded from

nucleoli indicates that the protein has low affinity for chromatin (the chromatin density in nucleoli is reduced). The other factors studied here bind stronger to chromatin and thus correlate better with chromatin density, leading to their depletion from nucleoli. The lower figure panel shows cells transiently transfected with GFP-MECP2. Since not every cell is transfected there are cells that do not express MECP2 but are stained with DAPI. In the revised Fig. 1, we now focused on our novel findings, namely the quantitation of the abundance of the different proteins in PCH. Furthermore, we included a graphical representation of our workflow that is intended to clarify the scope and the relation between the different measurements we conducted.

From the ChIP-seq experiments, the authors conclude "...the simultaneous presence of HP1, H3K9me3 and 5meC" (p.12). While logical and likely correct, it cannot be postulated from population based experiments. ChIP-re-chip might solve this issue.

We agree that we cannot make statements about the simultaneous presence of different factors on chromatin without Re-ChIP experiments. We clarified this in the revised manuscript.

"The cell cycle experiments are incomplete and have only been conducted on a fraction of the factors. For example, the authors state: "Localization of the DNA methyltransferase Dnmt1 to PCH was cell cycle-dependent" - Where is it shown?"

The cell cycle-dependent localization of many factors has already been published. Thus, we refrain from adding additional data on this topic. In particular, the localization of DNMT1 throughout the cell cycle has been studied and is referenced (Easwaran et al, 2004).

"The authors use PCH and satellite repeats interchangeably, but satellites are just a part of the story. "

We revised the manuscript to be more precise about the two terms. In ChIP experiments, our findings refer to the reads that map on the major satellite sequence whereas in microscopy experiments we identify PCH foci via the increased chromatin density. These foci include the satellite repeats as well as flanking regions and potentially other sequences.

"The authors predict sparse binding of Suv39h every ~170 nucleosomes (p.13-14). I wonder whether this prediction is testable."

The ratio between bound SUV39H and nucleosomes is no prediction but was measured by fluorescence microscopy-based experiments. In particular, we quantified the absolute amount of immobilized GFP-tagged SuUV39H in PCH foci and compared it to the amount of endogenous SUV39H and the known concentration of nucleosomes in PCH, yielding the ratio of SUV39H to nucleosomes in PCH. Unfortunately, we cannot make statements about the spatial distribution of SUV39H molecules since the underlying sequence is repetitive and therefore the binding pattern within this region cannot be assessed by deep sequencing.

Minor: Expanded view pages 33-37 are in a wrong order; P.5 line 19: H4K20 and Suv4-20h are not shown in Figure 1A but in Figure E1

We corrected the order of the respective pages as well as the figure reference.

Reviewer #3

"In this manuscript the authors characterized several quantitative parameters of factors enriched at pericentric heterochromatin domains in mouse cells in order to define a model, which describes how these factors contribute to the stability and confinement/compaction of these domains. These factors include proteins, DNA methylation and histone marks with emphasis on the H3K9me3 modification, which is the hallmark of these heterochromatin domains.

They used a combination of advanced microscopy techniques including imaging, FRAP and FCS to determine dynamics and concentration of these factors at pericentric domains. They also used available data from the literature and analytical ultracentrifugation to investigate HP1 dimerization dissociation constant.

They then integrated all these parameters to generate a model according to which, sparse nucleation sites containing the H3K9 HKMT Suv39H propagates by looping the H3K9me3 modification to nucleosomes in close spatial proximity to maintain pericentric heterochromatin status.

Taken together this manuscript represents a significant amount of work, which proposes a model that should be of interest for the scientific community in the fields of nuclear organization and

genome stability. However there are some points that should be considered (1) to improve the clarity of the manuscript, and (2) to cite some original work that reported same findings earlier than those described in this manuscript. Indeed several times in the manuscript, it is not clear for a non-specialist reader to discriminate novel findings from confirmation of already reported data. The added value of the work reported here is the modeling aspect and the characterization of numerical value for protein concentration/enrichment or association constant. Authors should thus concentrate on this aspect.

We thank this reviewer for appreciating our work. We restructured the Results and Discussion sections to clarify which findings are novel. Further, we focused on the quantitation of PCH at the beginning of the manuscript.

Below are some specific comments that authors should consider

1) *Some of the data presented in the manuscript are known for long time. For example the description of accumulation of HP1H3K9me3, Suv39h, MeCP2 and HP1 at pericentric domains in figure 1A along with the corresponding text could be shortened.*

The text was revised and the quantitative aspects of our work were strengthened. We further included a scheme of the workflow illustrating our quantitative analysis of the abundance and interactions of important PCH factors in the revised Fig. 1A.

2) *The fact that in Suv39h dn cells transcription of satellite sequences increase was published (Lenhertz et al., 2003 Current Biol.; Martens et al., 2003 Embo Journal)"*

These references are now included in the manuscript.

3) *"Decondensation of chromocenters upon deacetylase inhibitor was reported in (Taddei et al., 2001 Nat Cell Biol.)"*

This reference is now included in the manuscript.

4) *"Dynamics of HP1 determined by FRAP analysis was investigated in Festenstein et al., 2003, Science; Cheutin et al., 2003 Science; Schmiederberg et al., 2004 Molecular Biology of the Cell ; Dialynas et al., 2007 Journal of Cell Science. In this respect authors should discuss their own results in light of those published."*

We included the respective references in the revised manuscript. We want to point out that the FRAP analysis in these previous studies was conducted differently than our analysis. In particular, it turned out in recent years that reaction-diffusion processes have to be treated explicitly to obtain correct results (Müller et al, 2009; Sprague et al, 2004). Further, we think that it is important to conduct and analyze experiments in the same way in order to robustly compare the mobility of different proteins to each other.

5) *"Presence of Suv39H1 on mitotic chromosome was also reported in Melcher et al., Mol. Cell. Biol."*

We included the reference in the revised manuscript.

6) *"It is not clear whether every data reported here have been generated with cell lines that stably express the fluorophores or cell lines that are transiently transfected. This should be indicated in the figure legend for every figure showing quantification of enrichment and immobilized protein."*

We revised the figure legends accordingly to include this information.

References

Angel A, Song J, Dean C, Howard M (2011) A Polycomb-based switch underlying quantitative epigenetic memory. *Nature* **476**: 105-108

Berger SL, Kouzarides T, Shiekhhattar R, Shilatifard A (2009) An operational definition of epigenetics. *Genes Dev* **23**: 781-783

Bonasio R, Tu S, Reinberg D (2010) Molecular signals of epigenetic states. *Science* **330**: 612-616

Comet I, Schuettengruber B, Sexton T, Cavalli G (2011) A chromatin insulator driving three-dimensional Polycomb response element (PRE) contacts and Polycomb association with the chromatin fiber. *Proc Natl Acad Sci U S A* **108**: 2294-2299

Easwaran HP, Schermelleh L, Leonhardt H, Cardoso MC (2004) Replication-independent chromatin loading of Dnmt1 during G2 and M phases. *EMBO Rep* **5**: 1181-1186

Erdel F, Müller-Ott K, Rippe K (2013) Establishing epigenetic domains via chromatin-bound histone modifiers. *Ann N Y Acad Sci* **1305**: 29-43

Lehnertz B, Ueda Y, Derijck AA, Braunschweig U, Perez-Burgos L, Kubicek S, Chen T, Li E, Jenuwein T, Peters AH (2003) Suv39h-mediated histone H3 lysine 9 methylation directs DNA methylation to major satellite repeats at pericentric heterochromatin. *Curr Biol* **13**: 1192-1200

Martens JH, O'Sullivan RJ, Braunschweig U, Opravil S, Radolf M, Steinlein P, Jenuwein T (2005) The profile of repeat-associated histone lysine methylation states in the mouse epigenome. *EMBO J* **24**: 800-812

Müller KP, Erdel F, Caudron-Herger M, Marth C, Fodor BD, Richter M, Scaranaro M, Beaudouin J, Wachsmuth M, Rippe K (2009) Multiscale analysis of dynamics and interactions of heterochromatin protein 1 by fluorescence fluctuation microscopy. *Biophys J* **97**: 2876-2885

Probst AV, Almouzni G (2008) Pericentric heterochromatin: dynamic organization during early development in mammals. *Differentiation* **76**: 15-23

Schwartz YB, Linder-Basso D, Kharchenko PV, Tolstorukov MY, Kim M, Li HB, Gorchakov AA, Minoda A, Shanower G, Alekseyenko AA, Riddle NC, Jung YL, Gu T, Plachetka A, Elgin SC, Kuroda MI, Park PJ, Savitsky M, Karpen GH, Pirrotta V (2012) Nature and function of insulator protein binding sites in the Drosophila genome. *Genome Res* **22**: 2188-2198

Sprague BL, Pego RL, Stavreva DA, McNally JG (2004) Analysis of binding reactions by fluorescence recovery after photobleaching. *Biophys J* **86**: 3473-3495

Teif VB, Beshnova DA, Vainshtein Y, Marth C, Mallm JP, Hofer T, Rippe K (2014) Nucleosome repositioning links DNA (de)methylation and differential CTCF binding during stem cell development. *Genome Res*: published online 8 May 2014, doi 10.1101/gr.164418.164113

Van Bortle K, Ramos E, Takenaka N, Yang J, Wahi JE, Corces VG (2012) Drosophila CTCF tandemly aligns with other insulator proteins at the borders of H3K27me3 domains. *Genome Res* **22**: 2176-2187

2nd Editorial Decision

20 June 2014

Thank you again for submitting your work to Molecular Systems Biology. We have now heard back from the two referees who were asked to evaluate your manuscript. As you will see from the reports below, while their main concerns have been satisfactorily addressed, the reviewers mention two issues that we would ask you to address by providing additional comments/discussion in a revision of the manuscript.

On a more editorial level, we would like to draw your attention to the following:

- The mathematical model described in the study, should be provided in a machine-readable form and preferably, SBML format should be used instead of scripts (e.g. MATLAB). We strongly encourage deposition of models in a public database such as Biocompare (<http://biocompare.net/>). For further information on our policy regarding the availability of materials, data and software you can refer to our author guidelines (<<http://msb.embopress.org/authorguide#a2.4>>).
- We would like to ask you to include the GEO accession number for the ChIP-seq data (as mentioned in your last message).

Reviewer #1:

The authors have significantly improved the clarity of the ms and I certainly understand their conclusion better. However, I still feel that I do not understand the fundamental basis for the epigenetic memory in their system. The authors have pointed out in their rebuttal that the expression of major satellite can be different for different mouse cell types, with these states being heritable through mitosis. Indeed this behavior is epigenetic. They then emphasise that a high CpG density nor high level of 5meC are sufficient for H3K9me3-based silencing. So what is the fundamental difference between the cell types? Is it simply that the expression of the critical proteins (MECP2 for example) varies and is set at an epigenetically stable level. In that case the fundamental cause of the epigenetic memory would be a stable level of a trans-factor. Some insight into this, even if only to say that the answer is unknown, would be very useful.

Reviewer #2:

The authors addressed most of the criticism but there is one major point which was not explained properly. In my initial review I wrote: "The major caveat though is the fact that while the model nicely explains many of the PCH features, including, importantly, robustness and spatial confinement, it does not take into consideration and hence does not explain the extensive use of insulators in the mammalian genome".

In their answer, the authors write "...the wide-spread use of this model does not guarantee its correctness...". The original criticism did not refer to the wide use of the model in the literature, which is, I agree, irrelevant, but to the actual wide use of insulators in the mammalian genome! This point should be explained. How do the model deal with the the abundance of CTCF and other insulating proteins in the mammalian genome?

2nd Revision - authors' response

10 July 2014

Editorial requests:

"The mathematical model described in the study, should be provided in a machine-readable form and preferably, SBML format should be used instead of scripts (e.g. MATLAB). We strongly encourage deposition of models in a public database such as Biomodels (<http://biomodels.net/>)."

We have consulted an SBML developer and were advised that spatial features are not supported by the current standard. We modeled chromatin as a chain of 300 nucleosomes connected by DNA. The spatial order and relationship between nucleosomes at different positions on this chain is an essential component of our model (in terms of enumerating nucleosomes along the chromatin fiber and using interaction kernels). This feature cannot be implemented in the SBML format, which is designed to represent reactions between soluble factors that adopt a random spatial relation as long as they are in the same compartment. Accordingly, we do not see another option than providing the Mathematica script used for our computations to describe the model in machine-readable form.

"We would like to ask you to include the GEO accession number for the ChIP-seq data (as mentioned in your last message)."

The GEO accession numbers for the ChIP-seq sequencing data are now given in the manuscript

"Unfortunately, we are currently experiencing some technical issues that have resulted in delaying the implementation of the Expanded View format. Therefore, we are still publishing currently accepted manuscripts using the previous "Supplementary Information" Format."

We have changed the references to Supplementary Files into "Supplementary Figure/Table SX", in the Supplementary Information file and in the main text.

Reviewer #1

“The authors have significantly improved the clarity of the ms and I certainly understand their conclusion better. However, I still feel that I do not understand the fundamental basis for the epigenetic memory in their system. The authors have pointed out in their rebuttal that the expression of a major satellite can be different for different mouse cell types, with these states being heritable through mitosis. Indeed this behavior is epigenetic. They then emphasise that a high CpG density nor high level of 5meC are sufficient for H3K9me3-based silencing. So what is the fundamental difference between the cell types? Is it simply that the expression of the critical proteins (MECP2 for example) varies and is set at an epigenetically stable level. In that case the fundamental cause of the epigenetic memory would be a stable level of a trans-factor. Some insight into this, even if only to say that the answer is unknown, would be very useful.”

We are glad that we could clarify the issues raised previously. The remaining question about the origin of the differences of the pericentric heterochromatin (PCH) state between for example mouse embryonic stem cells (ESCs) and fibroblasts is very justified and relevant. As anticipated by the reviewer the underlying mechanisms are hardly understood and the suggestion that a critical protein varies in its concentration, which is set epigenetically to a cell-type specific stable level seems likely. A factor modulated in this manner could be the concentration of SUV39H1 and SUV39H2 enzymes. In ESCs there is two-fold less SUV39H1/2 than in NIH 3T3 fibroblasts as estimated on their normalized RNA expression levels. To illustrate this effect we have calculated from our model that a two-fold increase or reduction of the protein concentration of SUV39H (as the sum of SUV39H1 and SUV39H2) would raise or lower the H3K9me3 level in PCH from an average value of 38% to about 51% or 28%, respectively. This dependency is now shown as an additional plot in Fig. 8F. This figure panel replaces the previous Fig. 8E. In addition, SUV39H activity could also be differentially regulated by posttranslational modification like acetylation (Vaquero et al, 2007), methylation (Wang et al, 2013) or phosphorylation (Park et al, 2014). We have now discussed these issues in the revised manuscript in the Discussion on page 18, first paragraph.

Reviewer #2

“The authors addressed most of the criticism but there is one major point which was not explained properly. In my initial review I wrote: “The major caveat though is the fact that while the model nicely explains many of the PCH features, including, importantly, robustness and spatial confinement, it does not take into consideration and hence does not explain the extensive use of insulators in the mammalian genome”. In their answer, the authors write “...the wide-spread use of this model does not guarantee its correctness...”. The original criticism did not refer to the wide use of the model in the literature, which is, I agree, irrelevant, but to the actual wide use of insulators in the mammalian genome! This point should be explained. How do the model deal with the abundance of CTCF and other insulating proteins in the mammalian genome?”

We are glad to learn that we could address the criticism regarding our initial manuscript and apologize if we have misunderstood the point related to the abundance of insulators like CTCF. This issue can be clarified by considering that the designation of CTCF and related proteins as boundary or insulator factors refers to two different types of activities: One is it to limit the one-dimensional spreading of a chromatin state/modification along the nucleosome chain. As discussed in our manuscript such an activity is not required in the model we propose for pericentric heterochromatin. Additionally, we have doubts whether such a mechanism could robustly partition the genome, a view that is corroborated by the CTCF knockdown/knockout experiments conducted in previous studies (Schwartz et al, 2012; Van Bortle et al, 2012). The second activity that has been previously used to define an insulator activity of CTCF is it that long-range interactions of enhancers with their target gene are confined to a specific promoter and do not occur with another promoter on the same chromosome, which in some instances would be less distant on the chain. Such an activity can be rationalized with the ability of CTCF to induce the formation and stabilization of chromatin loops. For this function there is ample evidence in the literature as reviewed for example by Corces and coworkers who termed CTCF the “master weaver of the genome” (Ong & Corces, 2014). We have now clarified in the revised manuscript in the discussion on page 23, first paragraph that we do not question the existence of this type of boundary/insulator function by CTCF and other factors that originates from the ability to affect the higher-order folding of chromatin as an architectural factor. It is noted that such an activity would be fully consistent with directing epigenetic modifiers to certain

genomic loci (while preventing their interactions with others) in 3-dimensions via the ‘nucleation and looping’ mechanism that we propose in our study.

References

Ong CT, Corces VG (2014) CTCF: an architectural protein bridging genome topology and function. *Nat Rev Genet* **15**: 234-246

Park SH, Yu SE, Chai YG, Jang YK (2014) CDK2-dependent phosphorylation of Suv39H1 is involved in control of heterochromatin replication during cell cycle progression. *Nucleic Acids Res* **42**: 6196-6207

Schwartz YB, Linder-Basso D, Kharchenko PV, Tolstorukov MY, Kim M, Li HB, Gorchakov AA, Minoda A, Shanower G, Alekseyenko AA, Riddle NC, Jung YL, Gu T, Plachetka A, Elgin SC, Kuroda MI, Park PJ, Savitsky M, Karpen GH, Pirrotta V (2012) Nature and function of insulator protein binding sites in the Drosophila genome. *Genome Res* **22**: 2188-2198

Van Bortle K, Ramos E, Takenaka N, Yang J, Wahi JE, Corces VG (2012) Drosophila CTCF tandemly aligns with other insulator proteins at the borders of H3K27me3 domains. *Genome Res* **22**: 2176-2187

Vaquero A, Scher M, Erdjument-Bromage H, Tempst P, Serrano L, Reinberg D (2007) SIRT1 regulates the histone methyl-transferase SUV39H1 during heterochromatin formation. *Nature* **450**: 440-444

Wang D, Zhou J, Liu X, Lu D, Shen C, Du Y, Wei FZ, Song B, Lu X, Yu Y, Wang L, Zhao Y, Wang H, Yang Y, Akiyama Y, Zhang H, Zhu WG (2013) Methylation of SUV39H1 by SET7/9 results in heterochromatin relaxation and genome instability. *Proc Natl Acad Sci U S A* **110**: 5516-5521